# Towards Practical Tool Usage for Continually Learning Large Language Models

## Abstract

Large language models (LLMs) have demonstrated an innate ability to solve complex language-based tasks. Nevertheless, additional insights have suggested that they lack the capacity to adjust for either stored information or task-solving skills becoming outdated, as their knowledge, stored directly within their parameters, remains static in time after pre-training. As a way to counter this, tool use can help by offloading some of this work to systems which LLMs can access through an interface. Yet LLMs that use these may still need to adapt to nonstationary environments, as new tools can emerge and existing tools can change. Nevertheless, tools require less specialized knowledge, leading us to hypothesize that they may inherently be better suited for continual learning (CL) as they rely less on parametric memory for directly solving tasks and instead focus on learning when to apply pre-defined tools. To verify this, we develop synthetic arithmetic benchmarks and follow this by aggregating existing NLP tasks to form a more realistic testing scenario. While we demonstrate scaling model size is not an explicit solution to solving continual learning, regardless of tool usage, continual learning techniques can enable tool-augmented LLMs to both adapt faster while forgetting less, highlighting their potential as continual learners.

## 1 Introduction

The evaluation of large language models (Raffel et al., 2020; Chung et al., 2022; Touvron et al., 2023) have shown them to be highly performant on a variety of domains that appear knowledge intensive (Srivastava et al., 2023; OpenAI, 2023). Through the probing of model parameters (Petroni et al., 2021), a prevailing understanding of this is comes from the belief that LLMs possess a compressed representation of knowledge in their parameters, which can then be retrieved through language-based prompts. However, this knowledge is static unless directly intervened upon, which can lead to a lack of long-term potential. Existing knowledge can expire at different rates—*What is the current population of USA?* becomes obsolete in a decade while *Who is the President of X* expires in expectation around every 4 years. Thus models must learn to adjust specific facets of it's knowledge at various different moments. Similarly, models may be used in different scenarios where its specific goals may change and require the use of the knowledge space to adapt with these shifts. For example, being tasked as a calculator-like tool requires using parameters to emulate arithmetic operations, whereas acting as a translation tool instead allocates them for better understanding the underlying linguistic components of a phrase. While a model can perhaps perform each task individually, directly applying a model for a task on which it has no expertise is often ineffective.

Thus model performance can greatly vary in settings where the ground-truths or objectives change over time. Part of this stems directly from information being stored as *parametric knowledge* (Petroni et al., 2019), where the model must retrieve and utilize it correctly when prompted (Roberts et al., 2020). When the truth changes, then the model must learn to adapt its parametric knowledge with it. Additionally, even if the information within the world does not change, the environment may change in such a way that the goal of the LLM changes (Kenton et al., 2021). Hence the consensus is that the generated responses from pre-trained LLMs can become unreliable (Zhang & Choi, 2021; Komeili et al., 2022) due to knowledge becoming stale, and the LLM must learn to adapt to the new task or information in order to make its generated texts correct or relevant. The vanilla approach to avoid staleness is to collect more data that

better reflects the current world and re-train from scratch (Gao et al., 2020). Yet this comes with the major disadvantage is that the necessary resources grow with the data and since models store information directly within parameters, additional parameters are needed to hold the new knowledge (Jang et al., 2022), leading to an unsustainable cycle due to the limited resources that exist in the physical world.

Two popular alternative solutions have been pursued as a potential solution: One—*knowledge editing* (De Cao et al., 2021)— bases itself on the assumption that knowledge in LLMs' parameters can be updated by modifying the parameters directly. But editing factual knowledge can warp the innate knowledge structure of LLMs (Gupta et al., 2023) and approaches that do not directly intervene on the parameters require the use of additional memory (Mitchell et al., 2022b; Dong et al., 2022). Another is the usage of low-rank adapters (Hu et al., 2022), which freezes a base model and introduces smaller *adapters* which can be used to fine-tune the model for down-stream tasks without needing to train it explicitly. However, adapters are task specific, meaning this can be costly once the number of tasks has grown, and it is the adapter that is tasked with handling changes in the data rather than the model itself.

Tangential to the knowledge forgetting problem, LLMs are have recently been trained to use tools (Schick et al., 2023) through APIs that can either retrieve information from outside sources rather than parameters directly (Lewis et al., 2020) or which can offload more complex but deterministic reasoning to domain-specific external modules. This enables for the storage of information outside of the LLM, allowing for independent updates of the information which do not affect the model directly while the model only requires maintaining knowledge of the API interface to remain up-to-date. Though this provides a reasonable simplification to the differential expiry rates in knowledge or the shifts from changing objectives, tool-use itself does not make LLMs everlasting as both the tools themselves and the set of existing tools can change. Therefore LLMs which use them must also adapt. As such, tool-use itself is insufficient for the non-stationary setups as discussed in the **continual learning (CL)** literature (Ring, 1998; Thrun, 1998), where it is the model that must learn to autonomously adapt to change in either the state of the world as well as down-stream tasks. Within this setting, this points at the non-stationarity in the tool definition which can inherently lead to difficulties adjusting to distribution shifts, as learned features for specific tasks often cannot adapt to new ones (Kumar et al., 2022).

Such simplification of complex tasks also runs the risk of overfitting to present tasks, leading to forgetting the past (McCloskey & Cohen, 1989a; French, 1993; Xie et al., 2021) by large parameteric models. A careful treatment is therefore needed to modify the static knowledge repository of LLMs into models capable of continually adapting to the non-stationarity involved in learning tools that vary in complexity. We summarize our work as follows:

1. We propose a synthetic arithmetic dataset with Easy and Difficult splits, and benchmark LLMs of size 125M-13B on using the tools in a task of continual API learning.

2. We show that even with scale, LLMs are incapable of naively adapting to task shifts through sequential fine-tuning highlighting the drawback of mere parametric knowledge to handle distribution shifts. This persists even when LLMs are tasked with using tools rather than drawing directly from its knowledge.

3. However, with a replay buffer, we demonstrate that tool-augmented LLMs can adapt to these task shifts, whereas standard LLMs still fall short.

## 2 Related Works

**LLMs as Continual Learners.** Learning in a non-stationary setting has been treated formally in the continual learning (Chen & Liu, 2018) paradigm. The objective of continual learning (Thrun, 1998; Kirkpatrick et al., 2017) is to learn from a sequence of tasks without the forgetting (French, 1993) of previously seen tasks. With growing emphasis on language based applications, continual learning in training of LLMs has focused on two main directions:

(1) Task learning, where LLMs must learn multiple downstream tasks in sequence (Huang et al., 2021; Mehta et al., 2023).

(2) Domain adaptation, where the LLM is trained on multiple data domains (Gururangan et al., 2020; Ke et al., 2023) and must remain knowledgeable about each.

However, LLMs with large parameteric spaces limit the applicability of regularization-based techniques (Li & Hoiem, 2018; Lopez-Paz & Ranzato, 2017; Zenke et al., 2017; Aljundi et al., 2018) while the few-shot abilities of LLMs (Brown et al., 2020) suggest accommodating replay buffers (Rebuffi et al., 2017; Lopez-Paz & Ranzato, 2017; Shin et al., 2017; Chaudhry et al., 2019a; Wang et al., 2019b) of intractable sizes.

**Efficiently Updating LLMs.** Because LLMs are so costly to train (Strubell et al., 2019), there is great interest in keeping them up-to-date, which requires being able to update knowledge cheaply (Zhang et al., 2024). Within this space, two types of methods, parameter-preserving and parameter-editing, have emerged. Parameter-preserving methods, focus on keeping the underlying model intact (Dong et al., 2022; Huang et al., 2023; Hartvigsen et al., 2023; Zhong et al., 2023). Additional parameters or memory to track stale facts could quickly become impractical as the number of edits increases. Alternatively, parameter-editing methods directly modify the model parameters through fine-tuning the model to update only a select set of parameters (Zhu et al., 2021; Lee et al., 2022), meta-learning the parameters to edit (Mitchell et al., 2022a), or locating and modifying the relevant parameters (Santurkar et al., 2021; Tanno et al., 2022). This results in fast edits with little to no memory overhead. Yet the complicated structure of LLMs makes this a risky proposition, as modifying even one parameter can have various unknown downstream effects that can affect the usability of the model (Chen et al., 2023).

**Tool-Augmented LLMs.** LLMs are generalist agents that can be adapted to perform on a wide range of natural language tasks (Brown et al., 2020; Chowdhery et al., 2022). However, they still struggle in specialized settings (Patel et al., 2021; Lin et al., 2022) and have issues disassociating entities from extra-linguistic (Zhang & Choi, 2021) or even spurious (Joshi et al., 2022) contexts.

Tool-augmented LLMs (Schick et al., 2023) address this by learning to manipulate *specialized tools* to handle the knowledge-based computations that potentially fall outside the scope of the LLMs knowledge space. Wang et al. (2022); Imani et al. (2023); Paranjape et al. (2023) have shown improved zero-shot performance across a variety of downstream tasks without drops in language modeling abilities. Tools simplify tasks for LLMs, potentially reducing solving a task to learning to route to appropriate tools. However, these prior works do not study how tool-augmented LLMs adapt to new tasks or settings.

This work attempts to measure the issues that stem from LLMs forgetting by directly learning sequentially through the task samples. By replacing direct-learning with learning with tools, the work reposes the tasks in the tool space, and solves a unified non-stationarity problem of continual learning of tools as a proxy to solve the challenge of continual learning using task samples directly.

## 3 Motivation

### 3.1 Tools Usage in LLMs

**What are tools?** Wang et al. (2024) define an **LLM-used tool** as "*a function interface to a computer program that runs externally to the LLM, wher ethe LLM generates the function calls and input arguemnts in order to use the tool*". Similar to human-used tools, such tools are external to the employer (the entity who uses the tool) while remaining part of the environment (Shumaker et al., 2011). Though tools exist for many use cases, they are meant to facilitate the task-solving process for the employer by abstracting parts of the process or by enhancing the employer with new abilities to lessen the burden of solving it.

**Why are tools helfpul/useful?** Much like with human tool usage, tools can help LLMs solve tasks in a variety of different ways. For example, they can help the LLM through providing additional information for making calculated decisions. Additionally, they can also help through the ability to make specific computations more efficient or more precise. However, despite how they can be useful for various purposes, there also exist scenarios where their use can be redundant or even potentially harmful. More specifically, if the tools is meant to simply emulate part of the LLM they are meant to help, the tool can be considered useless

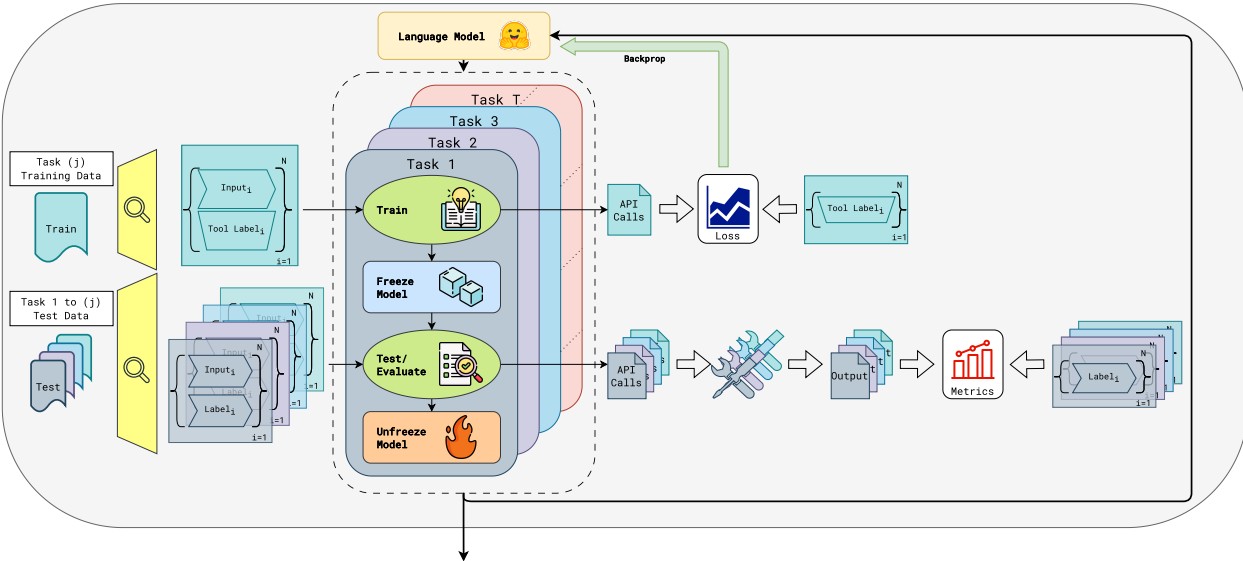

Figure 1: Our Continual Learning with Tools Setup. For a task, the model is first trained to predict/generate tool calls, rather than explicit responses. The trained model is then evaluated on all tasks on which it has already been trained, during which it outputs tool calls that are parsed and executed to return an output which is compared against the ground-truth. The model is then trained on the next task in the sequence. This is repeated until all tasks have been learned by the model.

in nature. Another scenario is if the tool, despite its benefits from a computational perspective, provides fewer guarantees compared to the LLM itself, potentially rending it less effective when used in practice.

**The tool use paradigm.** The standard tool-usage paradigm can be considered a setting where a user communicates with the language model in natural language text, which the language model has to then process either itself or with the use of tools. When the use of external tools is necessary to complete the task, e.g., get real-time weather information, the language model can generates tokens of the tool name and corresponding input arguments enclosed with (parentheses) to construct a complete tool calling expression. This expression can then be evaluated using a server to complete the expression with the tool-specific output and then return the execution result to the LLM.

## 3.2 Motivating Questions

Continual learning problems motivate the need for more robust models or networks, as drops in performance can severely limit their practical applicability in various settings, particularly ones that have shifts in the model usage or training data over time. LLMs are no exception to this, as the changing environment in which it is deployed can often mean that the knowledge it already possesses might need to be adapted with time or replaced when it becomes irrelevant or undesired. More formally, continually adapting LLMs to the changing world and domain knowledge is a complex but relevant problem, as forgetting prior information can limit the applicability of LLMs. Further, with shifts in domain being aperiodic for diverse knowledge and LLMs being the generalist model they are leads us to the pertinent question:

> *Can learning to use tools alleviate sequential learning challenges?*

and the sub-questions that need to be answered:

- (Q1) *How far can we push by simply increasing parametric knowledge space help for continual learning?*
- (Q2) *Are there limits to how much both tool-augmented LLMs and vanilla LLMs can learn continually?*

(Q3) *How do tool-augmented LLMs fare with imperfect tools?*

We use these questions to build our methodology and experimental design in the following sections.

## 4 Methodology

### 4.1 Preliminaries

**Model:** We use auto-regressive Transformer-based language models in a text-generation setup, in particular the OPT (Zhang et al., 2022) family of pre-trained LLMs up to 13 billion parameters. This allows us to compare the abilities of similar generative language models along with how they change with scale.

**Dataset Format:** To properly assess the usefulness of tool learning, each sample $s \in \mathcal{D}$ consists of a query $s_q$, the raw answer to the query, $s_G$, and an API call answer $s_A$, which can be executed by a task-specific API to obtain a response that is compared with $s_G$ using exact string matching.

**Learning Setup:** Language models are trained either **with tools** or **without tools** to solve a sequence of $T$ tasks—$\{\mathcal{T}_1, \ldots, \mathcal{T}_T\}$. Each task $\mathcal{T}_k$ defines a specific tool and a dataset $\mathcal{D}_k$ which contains the examples associated with learning the $k$-th tool. **With tools**, the model learns to generate the API calls, as mentioned previously, that gets routed to the appropriate API to generate the answer. **Without tools**, the model is fine-tuned to predict the answer directly, such as a numerical or textual response. Iterating over tasks in sequence, at every iteration, $i$, a model is trained with examples corresponding to $\mathcal{T}_i$ and evaluated on test sets of all the tasks the model has seen until then. Each task uses a learning rate warm-up followed by a decay to 0, *i.e.* the learning rate warm-up and decay repeats for each task in the set. We use the AdamW (Loshchilov & Hutter, 2019) optimizer with a peak learning rate based on the model size[1].

### 4.2 Baselines

For each setup, we train under a number of settings:

**Sequential Fine-tuning:** The model sees a stream of tasks $\{\mathcal{T}_i\}_{i=1}^T$ in an order without repetition or explicit rehersal on examples from prior tasks. The model is explicitly fine-tuned on each task and once complete moves to the next task for training. This serves as a naive baseline for continual learning, as there is no explicit control that is being used to ensure that information is being retained between tasks.

**Mixed Dataset:** All tasks are mixed into a single task to train a model. This is equivalent to "seeing" all tasks at once and is a soft upper-bound, where model learns from all available data at once and task specific information is trivially being retained due to all examples being merged within a single dataset Because there is no task shift that is observed in these settings, accuracy is often reported as a single value over the entire dataset. However, we consider this a soft upper-bound as the model may still not solve the task completely under this setup and therefore it remains plausible that some other methods may be able to achieve greater performance even in a multi-task setup.

**Episodic Replay (ER):** Chaudhry et al. (2019b) augment models with a replay buffer that retains examples from the previous tasks. With the buffer, the model continually takes some of the recent data and randomly replaces older samples. When training, the model will randomly sample a batch from the replay buffer and calculate a replay loss which is added to the standard loss before performing a gradient update. Motivating the usage of this method are observations that LLMs are few-shot learners (Brown et al., 2020), suggesting that this may be an efficient use case of the method given the smaller number of examples and subsequent buffer size that may be necessary.

---

[1]Hyper-parameters are provided in Appendix C

### 4.3 Evaluation Metrics

We evaluate using the following metrics for measuring performance[2]:

**Accuracy**: Given that each task $\{\mathcal{T}_1, \ldots, \mathcal{T}_T\}$ consists of a train and test set, we can measure the accuracy on each test set individually. We report the average accuracy on test sets up to the most recent task on which the model was trained. In particular, suppose a model is being trained on task $\mathcal{T}_\tau$. The average accuracy is measured as in Equation 1,

$$\hat{p}_\tau = \frac{1}{\tau} \sum_{k=1}^{\tau} p_{\tau,k} \tag{1}$$

where $p_{\tau,k}$ denotes the performance of the model on the test set associated with task $k$ after having trained on task $\tau$. In the tool setup, $p_{\tau,k}$ is measured by parsing and executing the generated API calls and computed exact match (Rajpurkar et al., 2016) with the true answers.

**Forgetting** (Chaudhry et al., 2018): Forgetting is the average degradation ($\geq 0$) in performance on all seen tasks excluding the most recent task on which the model was trained. It is often defined as

$$f_\tau = \frac{1}{\tau - 1} \sum_{k=1}^{\tau-1} \max_{t \in \{1, \ldots \tau - 1\}} \left( \frac{p_{t,k} - p_{\tau,k}}{p_{t,k}}, 0 \right), \tag{2}$$

where $p_{t,k}$ is the performance on task $t$ that has been previously observed up to when it has been trained on task $k$. As such, Equation 2 measures the average forgetting a model has observed for all task on which it has already been trained (up to $\mathcal{T}_\tau$). Lower forgetting is desirable as it indicates that the knowledge necessary to solve prior tasks was not over-written while learning the more recent ones, or that the model has generalized in such a way that previous and incoming tasks can be solved with the same knowledge.

**Learning Accuracy** (Riemer et al., 2019): In many scenarios, models have limited capacity and can learn a limited number of tasks. This is determined both by the difficulty of the tasks that need to be learned, as well as the size and configuration of the model. However, in order for a model to be a strong continual learner, it is often necessary that they retain the ability to learn incoming tasks effectively.

$$L_\tau = \frac{1}{\tau} \sum_{k=1}^{\tau} p_{k,k}, \tag{3}$$

Equation 3 defines learning accuracy (L-A) to approximate the learning capacity by measuring average performance on each task immediately after being trained on it. Consistently high learning accuracy is important, as it indicates that the model has high flexibility towards learning new tasks that are introduced.

## 5 Experiments

We design our experiments to address Q1-3 in §3.2 as follows.

**Does more parametric space solve CL?** Understanding the effects of scale requires ensuring that larger models can adequately solve the task when not presented in a sequential learning setting. For Q1, we first construct a synthetic arithmetic dataset of functions (Table 1), each with a single template and limited to integers between 0 to 99. Every sample adheres to the format defined in §4. Each operation has an associated API format answer, ex. ADD($a$, $b$) where $a$ and $b$ are arguments provided to the tool, and explicit numerical answers. Non-integers are expressed in decimal format, e.g. 0.75. Tasks have the same number of examples, divided into training and test sets[3]. We refer to this as our *Toy Arithmetic Task*, in particular due to the relative simplicity of the task.

---

[2]Formulas and further details are provided in Appendix D.
[3]Additional hyper-parameters are found in Appendix C

With these, we verify if increasing the number of parameters of LLM improves performance, measured through with accuracy and forgetting. If accuracy increases while forgetting decreases with the parameters, then we can infer that link between the size of the parametric space and performance is in fact linked. However, as we later observe and discuss in §6, this is far from the case.

**How much can LLMs learn continually?** Following the observations meant to answer Q1, we further the study and attempt to look at the extent to which LLMs can learn continually as the setting become more difficult, both through the number of tasks or the general dissimilarities between them. As such, the next goal is to observe some of these limits in both with and without using tools by increasing the difficulty of the overall setting, for which we build a more difficult arithmetic benchmark.

| Description | API Format | Example |
|---|---|---|
| Add two numbers | ADD(A, B) | ADD(23, 35) = 58 |
| Subtract $b$ from $a$ | SUB(A, B) | SUB(34, 12) = 12 |
| Multiply two numbers | MULT(A, B) | MULT(5, 7) = 35 |
| Divide $a$ by $b$ | DIV(A, B) | DIV(81, 2) = 41.5 |
| Compute the GCD | GCD(A, B) | GCD(12, 20) = 4 |
| Compute the LCM | LCM(A, B) | LCM(14, 21) = 42 |
| List the prime factors | LP(A) | LP(4) = 1, 2 |

Table 1: Tasks within the toy setup (top) and additional operations in the advanced setup (bottom).

This is done through expanding the input/output space along with more templates. We create additional functions and templates for existing functions which must be learned to properly use the tools. Furthermore, we add more ambiguity to the tool templates by increasing the token similarity for many templates. The output space now covers operands which can include real numbers, hence requiring the models to be capable of identifying which parts of the input properly constitute parts of the tools calls and use this to create the output. We refer to this as our *Advanced Arithmetic Task.*

The goal here is to observe if performance (as measured in the same way in the previous task) can remain constant between the two tasks both with and without using the tools. We expect that although accuracy may drop in part due to the task difficulty, but does forgetting remain consistent between the toy task and this more difficult one? In particular, if forgetting is consistently greater for the advanced task, this highlights limitations in the LLM with respect to becoming more general, autonomous multi-task learners.

**Do tool-augmented LLMs behave well with imperfect tools?** While our arithmetic benchmarks are useful for evaluating LLMs in a continual learning setting, the goal remains to use them for more practical use cases. In such cases, oracles are not readily available in the same way as in the arithmetic settings. Explicit rules exist for arithmetic tasks; if one designs a tool to follow said rules, then mastering the tool is equivalent to mastering the task. But this assumption fails to hold in many cases, which motivates our next question: do the benefits of tool learning still hold once we move away from perfect tools. For this, we construct a more representative benchmark where the LLM attempts to learn tools that correspond to different natural language understanding tasks.

We use a subset of tasks from the GLUE benchmark (Wang et al., 2019a), in particular MNLI, QQP, SST-2 and CoLA, to be learnt continually. Each task assumes a specific tool. For example, samples from QQP ask if two questions are paraphrases of each other; we provide a tool response with a tool PARAPHRASE requiring two string arguments. For the tool response, we use the API answer as input to a separate language model which returns the final response as the tools are no longer oracles; they can produce an incorrect answer.

This aims to address Q3 by providing a more realistic scenario for continual learning where the tools themselves cannot completely solve the task. Here, we wish to verify if the gaps we observe in the arithmetic setups disappear simply through the use of imperfect tools, again through the lens of accuracy and forgetting.

## 6 Results and Analysis

### 6.1 LLMs struggle with Continual Learning, irrespective of tools

In Figure 2, we compare the performances of the differently sized models (with the same architechture) on both the synthetic arithmetic datasets and the realistic task as described in §5.

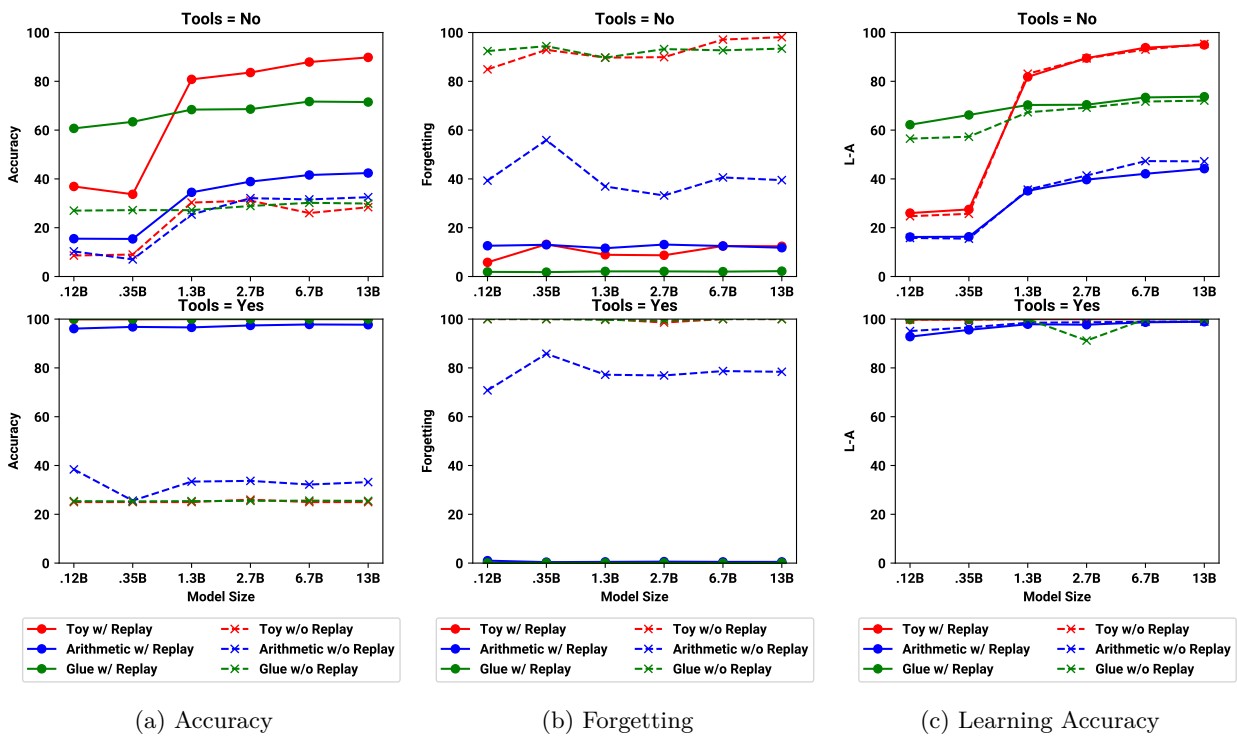

(a) Accuracy        (b) Forgetting        (c) Learning Accuracy

Figure 2: Across the different task setups, we measure the different metrics. Although it is evident that using tools improves the learning accuracy significantly, we observe that the Accuracy across tasks is not reflecting the same. The significant forgetting of tools only get fixed with appropriate usage of a replay buffer to improve the overall accuracy irrespective of the task difficulty.

Experimenting with both directly learning over the samples and using APIs, we observe that generalizing on arithmetic tasks is challenging when learning directly from raw samples (top plots in Figure 2). In particular, the model appears to overfit the final couple tasks, as indicated by an accuracy that plateaus even with increasing model size (Figure 2a) while forgetting appears to be near complete (Figure 2b). This persists even when we switch with tools (still without the use of a replay buffer). Furthermore, though the learning accuracy for a small language model using tools is higher than that of a $100\times$ larger model not using tools, the retention of past tasks as observed in the overall accuracy in Figure 2a appears as a prevalent issue across the model sizes, as accuracy plateaus while forgetting remains signfcant in all these cases.

While demonstrates that LLMs struggle with sequential learning in general, we look at whether the performance degradation is an artifact that comes with the learning set up. To this end, we compare the performances of the models in a mixed dataset setting where the models learn all the tasks at once

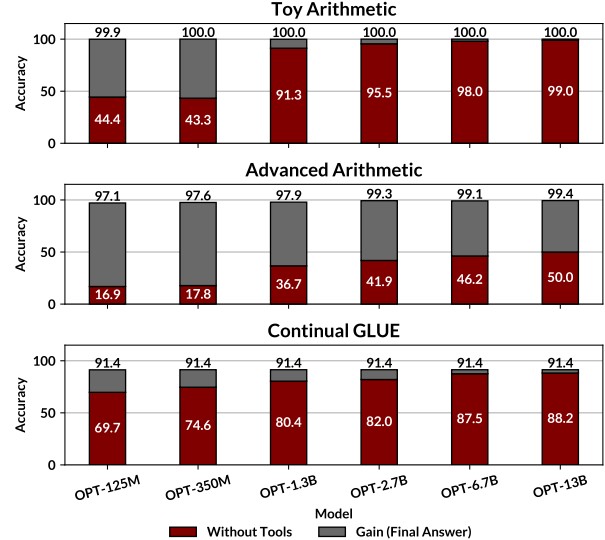

Figure 3: Accuracy on all benchmarks when tasks are mixed. Red bars note accuracy without tools, grey bars show the gain from using tools. Top labels show accuracy using tools.

with and without using the tools. The hypothesis is that if the LMs show significant retention as indicated

with the comparable performances to using tools, it can be regarded that more data potentially solves the forgetting problem. But, contrary to Figure 3, we observe that the gap does exist in the different tasks. Accordingly, irrespective of using tools or tasks seen all at once or not, LLMs struggle with generalizing and adapting to the task changes.

## 6.2 More parameters does not negate forgetting

A common question that arises in continual learning research is whether or not a larger parameter space can help mitigate forgetting in neural networks, with ambiguous conclusions (Ramasesh et al., 2022; Lee, 2024). Accordingly, we conduct an analysis of what occurs within our setting. Figure 4a indicate the effect of model size on the ability of learning tasks to increase with model size. However, from Figure 4b, we fail to see any systematic decrease in the forgetting of the model, suggesting that being able to learn tasks sequentially remains a concern despite the increase in model capacity. Nevertheless, the greater learning accuracy observed with larger models can be useful to unleash the potential of tool-augmented LLMs.

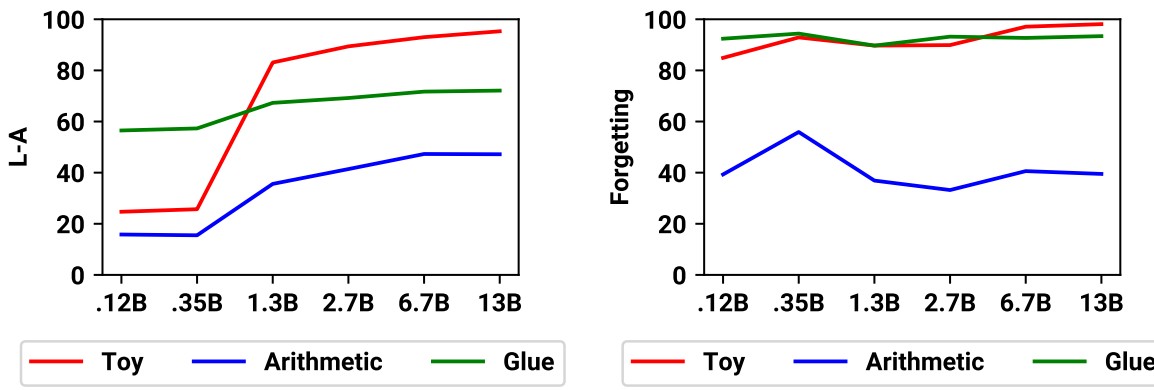

(a) Changes in Learning Accuracy as the model size increases. Learning Accuracy visibly increases with scale, suggesting that larger models can inherently adapt or overfit new tasks at a faster rate.

(b) Changes in Forgetting as the model size increases. A lack of a visible trend suggests that forgetting is no inherently the cause of model size, but inherent to how the model is trained or functions.

Figure 4: While we observe that the scale of the model (when not using tools) plays a significant role in how the model's capacity is used to learn an incoming task, the lack of similar effect with forgetting suggests a 13B model and a 125M model are equivelent in terms of in retaining knowledge of past tasks. This shows how despite scaling models, continually training them on different data can present a challenge due to the high levels of forgetting that occur with previously learnt information.

In particular, we observe in Figure 4 that tool-augmented LLMs' learning accuracy to be consistently higher than vanilla LLMs, suggesting a faster adaptation with tools. Even more encouraging is the fact that learning accuracy for the smallest tool-augmented LLMs is often far superior compared to the largest vanilla LLMs. This is promising, as it demonstrates that if one can overcome the forgetting concern that plagues LLMs in general, then using tool-augmented LLMs may be much more efficient than vanilla LLMs as they can replace ones that are larger for similar performance. This observation not only is evident when the tools are non-parametric oracles as in our arithmetic tasks but also in the case of our continual GLUE task where tools themselves are parametric models. Though models are no longer oracles, as demonstrated by imperfect learning accuracy (Figure 2c), the combined parametric space with smaller experts is still significantly smaller than a vanilla LLM that achieves equivalent performance.

By reposing problems in the tool space, models learn only to make the correct API calls and we see smaller models with tools to perform on par with larger models not using tools. Beyond a simplistic comparison, this could also be seen as an economic way to guarantee consistency and truthfulness to the results while not

incurring the cost of pre-training larger LLMs as the reliance is less on the more riskier LLMs' parametric knowledge (Kazemnejad et al., 2023).

These results motivate potential opportunities in building smaller models and learnable API calls that can outsmart large LLMs in terms of efficiency with cheaper training costs. While LLMs trained for more complex interaction and usage exist, such as instruction fine-tuned ones (Askell et al., 2021; Ouyang et al., 2022; Dubois et al., 2023), they still rely on the assumption that the underlying world does not change; one can still expect false statements unless they are explicitly trained to rely on outside data sources accessible in a predetermined manner. As such, tool-augmented LLMs present an opportunity to move away from larger models and towards smaller, more accessible ones with comparable use.

### 6.3 Tools can be beneficial for Continual Learning

By adopting more wide-spread techniques from continual learning, tool-augmented LLMs display significant advantages over prototypical LLMs. In particular, by using replay buffer, we observe that forgetting is alleviated to a significantly higher degree when learning with tools. In Figure 5, we observe that forgetting drops by $\geq 70\%$ in all tasks. By comparison, forgetting remains in the 5-15% range for arithmetic tasks and $\sim 2\%$ for the GLUE task when not using tools, which are all greater than $10\times$ the amount of forgetting that occurs with tools and replay. Though we observe that tool-augmented LLMs forget more than vanilla LLMs without replay, the amount of forgetting remains significant (over 80%, 30% and 85% for the three tasks) and limits their practical viability.

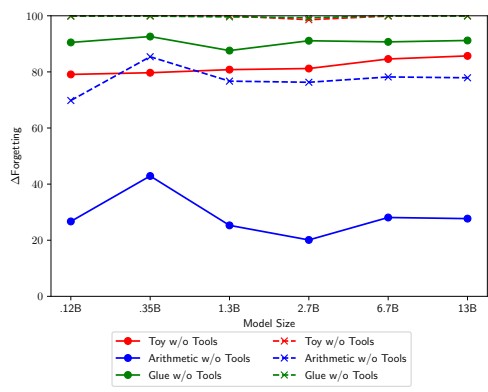

Figure 5: The replay buffer plays a significant role in aiding LMs across tasks in mitigating the forgetting. In particular, it reduces forgetting in all task settings both with and without tools.

What remains important, however, is that models appear capable of learning tools to a much greater capacity, shown by superior learning accuracy throughout. These benefits can be observed when using replay (solid lines in Figure 2), where we note the models learn to use the tools almost perfectly, and the tool LLM can significantly outperform vanilla LLMs in our arithmetic tasks. Even in the case of the more nuanced GLUE task, where the tool is not always correct, benefits are still visible as errors in the final answer result only from the imperfections with the tool, which we can remark due to the fact that the API call accuracy is perfect in these scenarios (see Appendix F). These observations bring us to hypothesize that through tool use, LLMs become better at utilizing their parametric knowledge, leading to greater task transfer during CL and allowing them to adapt more effectively.

## 7 Discussion

**Parametric Knowledge Utilization.** Studies into language models have shown that pre-training data is oftentimes directly available within trained parameters (Brown et al., 2020; Jiang et al., 2020; Qin & Eisner, 2021) as *parametric knowledge*. However, if the knowledge stored is very example dependent, then it is likely not usable Kazemnejad et al. (2023) in many instances, as there is no direct link between the context in which the knowledge was seen and other examples which are presented to the model (Prato et al., 2023). As such, one may question whether this knowledge space could be better used.

In contrast, tool learning can generalize the output space, as the learned samples can be more clearly separated into categories based on the tools that are used to solve them. This can make it easier to understand how to handle individual examples from the model perspective and maintain some memory of prior tasks. These observations can explain many of our results, such as improved learning accuracy but greater forgetting when learning tools without replay. If answers are all either numerical values or similar natural language

words, there possibly exists a smaller distribution shift that occurs when moving from one task to another. As a result, over-fitting to the answer format may result in a smaller performance degradation.

**Using Auxiliary Expert Systems.** Tool-augmented LLMs assume that the tools themselves are accurate for the task of interest as otherwise it's existence would be meaningless. But teaching LLMs to make use of tools as auxiliary systems remains a nuanced process; how does it know when to trust the system and take the system response as the truth? There is often a trade-off that exists between speed and performance in these cases; the faster we want the response to be then the more trust we must place in the system to be accurate and not double-guess it.

Tool-augmented LLMs can further be seen as an alternative to mixture of expert models (Jacobs et al., 1991; Shazeer et al., 2017; Fedus et al., 2022), which route examples to different experts. However, one can view Tool-augmented LLMs as a case where the expert exists externally; this leads to a system that may be less coupled with the task. Furthermore, introducing auxiliary systems bring about additional questions. For example, how do we ensure that the model can continuously maintain the ability to use the system properly? How is the knowledge for using tools stored and what does it inform us about how much the LLM knows about the tool? These require further analysis which are necessary both for practical use as well as for understanding LLMs in general.

**The Dual Nature of Forgetting.** Forgetting is a natural phenomenon, both in humans (Wang et al., 2020) and neural networks (French, 1999). While it is commonly agreed upon that a symbiotic relationship exists between learning and forgetting within humans (Bjork & Allen, 1970; Bjork & Bjork, 2019; Gravitz, 2019), forgetting is still treated as the cause of various failure modes within machine learning (McCloskey & Cohen, 1989b; Ratcliff, 1990). However works have began to show how forgetting and learning can work together symbiotically (Zhou et al., 2022).

In most contexts, forgetting is deemed a negative phenomena which hinders models. However, in the real world, this assessment may not always hold, especially when updating models with ease is important. For this, unnecessary information should be forgotten as quickly as new information is learnt. This means that forgetting is not a simple black-or-white issue, as when information can become out-dated or incorrect, forgetting it is desirable given that it is no longer useful. Therefore, tool-based models displaying higher forgetting but greater learning accuracy may in fact be desirable, as it demonstrates that models can maintain an ability to learn new information but simultaneously discard information that is no longer relevant.

## 8 Conclusion

In this work, we explore the potential use of tools in continual learning for LLMs. We apply this setup within a simple arithmetic reasoning setup, where a language model is taught multiple math functions. Our results demonstrate that LLMs that learning to generate answers based on tools both adapt faster to new tasks while also maintaining greater performance on prior tasks. We further validate these conclusions through a continual learning benchmark based on the GLUE natural language understanding benchmark. However, continual learning remains unsolved, as cases still exist where all models fail to demonstrate the ability to autonomously solve the benchmark. This emphasizes the need for models which can adapt to the world in the same manner as conscious humans and by highlighting current limitations and the potential for tool-augmented LLMs in this setting, these results hopefully delineate paths for future research which can allow for more practical LLMs deployed in the real world.

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

| Model | Toy | Advanced | GLUE |
|---|---|---|---|
| OPT-125M | 15 min | 30 min | 25 min |
| OPT-350M | 15 min | 30 min | 25 min |
| OPT-1.3B | 2 h | 4 h | 3 h 20 min |
| OPT-2.7B | 3 h | 6 h 30 min | 5 h |
| OPT-6.7B | 4 h | 8 min | 7 h 20 min |
| OPT-13B | 12 h | 24 h | 21 h 30 min |

Table 2: Approximate training times for each model.

## A    Implementation Details

For the experiments in this study, we exclusively use a server of 4 NVIDIA V100-32GB GPUs. Models with $\leq$ 350M parameters were trained on a single GPU. Other models used the entire server of 4 GPUs. The approximate training time (rounded to the nearest hour) for a single seed for each model without replay are presented in Table 2. Times with replay require approximately twice the amount of computation time.

## B    Benchmarks

### B.1    Dataset Size

**Toy Arithmetic Benchmark.** We generate an example for each function and operand. This results in 10000 examples per task, which we separate into a train and test set using an 80%-20% random split.

**Advanced Arithmetic Benchmark.** We generate 20000 examples per task, which we again separate into a train and test set using an 80%-20% random split.

**GLUE Benchmark.** Due to different tasks having different sizes, we use the size of the smallest task as our limits for both the train and test set for each task. In our case, we use 8.5k example from the training set of each task and 5k examples from the test set of each task.

### B.2    Templates

Templates for our benchmarks are in Table 3 and Table 4.

## C    Training Hyperparameters

We use the following hyperparameters for training a model on each task within the continual learning setup.

### C.1    Training Task Sequences

In continual learning, seeding occurs across the task sequence rather than the individual training examples per sequence. For example, given a set of tasks $\mathcal{T} = \{\mathcal{T}_1, \mathcal{T}_2, \ldots, \mathcal{T}_T\}$, the training can occur over any permutation of the task ordering.

| Description | Templates |
|---|---|
| Addition | What is $a$ plus $b$?
What is the sum of $a$ and $b$?
What do you get if you add $a$ to $b$?
What do the total of $a$ and $b$? |
| Subtraction | What is $a$ minus $b$?
What is the difference between $a$ and $b$?
What is $b$ less than $a$?
What is $a$ take away $b$?
What is the distance between $a$ and $b$? |
| Multiplication | What is $a$ times $b$?
What is the product of $a$ and $b$?
How much is $a$ groups of $b$?
$a$ multiples of $b$ is how much? |
| Division | What is $a$ divided by $b$?
What is the quotient of $a$ and $b$?
How many times does $b$ fit into $a$?
How $b$ go into $a$? |
| GCD | What is the greatest common factor of $a$ and $b$?
Calculate the highest common divisor of $a$ and $b$.
What is the largest number that divides both $a$ and $b$. |
| LCM | What is the smallest common multiple of $a$ and $b$?
What is the smallest number that is a multiple of both $a$ and $b$. |
| Prime Factors | What are the prime factors of $a$?
Which factors of $a$ are prime? |

Table 3: Templates for arithmetic tools.

| Task | Templates | Answers |
|---|---|---|
| MNLI | Does [Sentence 1] either entail or contradict [Sentence 2], or neither?
Is [Sentence 2] an entailment or contradiction of [Sentence 1], or neither? | {entailment, contradiction, neutral} |
| QQP | Is [Sentence 1] a paraphrasing of [Sentence 2]?
Are [Sentence 1] and [Sentence 2] paraphrases? | {yes, no} |
| CoLA | Is [Sentence 1] a linguistically acceptable sentence?
Does [Sentence 1] make sense linguistically? | {yes, no} |
| SST-2 | Does [Sentence 1] express a positive or negative sentiment?
What kind of sentiment does [Sentence 1] show? | {positive, negative} |

Table 4: Templates for GLUE tools.

# D   Evaluation Metrics

**Average Performance** We denote the performance on the test set of task $\tau$ when the model is being trained on task $T$ as $p_{T,\tau}$. The average performance across all learned tasks is therefore

$$\bar{p} = \frac{1}{T} \sum_{\tau=1}^{T} p_{T,\tau}$$

| Hyperparameter | Value |
|---|---|
| Batch Size | 64 |
| Peak Learning Rate | If (# of parameters $< 10^9$): $2 \times 10^{-5}$ 
 else: $1 \times 10^{-5}$ |
| Weight Decay | 0.01 |
| Adam $\epsilon$ | $1 \times 10^{-8}$ |
| Adam $\beta_1$ | 0.9 |
| Adam $\beta_2$ | 0.99 |
| Learning Rate Scheduler | Linear Warm-Up and Decay |
| Warm-Up (Per Task) | 10% of training steps |
| Gradient Clipping | 0.0 |

Table 5: Hyper-parameters for learning each tool.

**Forgetting (Chaudhry et al., 2018)** Supposing that the model is being trained on task $T$. Forgetting is defined as the amount by which performance on all previous tasks has degraded from it's previous maximum performance. If the performance metric is defined such that increased performance corresponds to an increase in metric value, then forgetting can be calculated as

$$F_T = \frac{1}{T-1} \sum_{\tau=1}^{T-1} \max_{t \in \{1, \dots T-1\}} (p_{t,\tau} - p_{T,\tau})$$

Otherwise, if the performance metric is defined such that increased performance corresponds to an increase in metric value, then forgetting can be calculated as

$$F_T = \frac{1}{T-1} \sum_{\tau=1}^{T-1} \min_{t \in \{1, \dots T-1\}} (p_{T,\tau} - p_{t,\tau})$$

However, this can converted to a relative forgetting over tasks

$$F_T = \frac{1}{T-1} \sum_{\tau=1}^{T-1} \max_{t \in \{1, \dots T-1\}} \left( \frac{p_{t,\tau} - p_{T,\tau}}{p_{t,\tau}} \right)$$

to account for general performance differences between tasks.

To further illustrate this, again suppose a sequence of task being presented to the model, $\{\mathcal{T}_1, \dots, \mathcal{T}_n\}$. The model has been trained on $\mathcal{T}_1$ and $\mathcal{T}_2$. If the model originally achieved an accuracy of 85% on $\mathcal{T}_1$ after training on $\mathcal{T}_1$ and this number drops to 50% after training on $\mathcal{T}_2$, we would say that there is a degradation of performance of 35%. However, in order to account for the fact that peak performance on tasks can differ, we normalize this value using the highest achieved accuracy on each task. As such, forgetting is presented as a normalized value between 0 and 1, representing the average degradation from peak performance across all tasks. This is only calculated from the second task onwards.

**Learning Capacity (Riemer et al., 2019)** Learning capacity is defined as the ability to learn a new task immediately as the model sees it. Therefore it can be defined as

$$L_T = \frac{1}{T} \sum_{\tau=1}^{T} p_{\tau,\tau}$$

where $p_{i,j}$ measures the accuracy on the test set of task $j$ while learning task $i$.

As an example, again suppose a sequence of tasks $\{\mathcal{T}_1, \dots, \mathcal{T}_n\}$ are presented to a model. After training on $\mathcal{T}_1$, we record the average performance on $\mathcal{T}_1$'s test set and report this as the learning accuracy. After training on $\mathcal{T}_2$, we record performance on the test set for $\mathcal{T}_2$, which we then average with the previously calculated immediate accuracy after $\mathcal{T}_1$ to record the learning accuracy after $\mathcal{T}_2$.

# E  Tools

## E.1  Evaluating API Calls

In order to evaluate API calls, we create a tool which can parse the generated calls. For example, if the generated call is 'ADD(5, 5)', then use a regular expression parser to separate 'ADD' as the function and '5, 5' as the arguments passed to the function. We then take this parsed answer, feed it to a pre-written function that can evaluate it and then compare it to the raw numerical answer.

## E.2  Building our tools

For our arithmetic tasks, we explicitly design functions which take integer arguments. As an example, we simply build our function for addition as

```
def add(a, b):
    return a + b
```

with the output being directly compared with the label provided in the dataset.

For the continual GLUE tasks, we instead use a frozen LLM available from HuggingFace that is meant to produce an answer, for example

```
from transformers import pipeline
def entailment(sentence, model=None):
    pipe = pipeline(model=model)
    return pipe(sentence)[0]['label']
```

after which we again compare the returned label with the ground truth provided in the dataset. By default, we use publicly available base-sized RoBERTa models that have already been fine-tuned on the corresponding task as the tool model.

# F  Detailed results

The following sections provide some additional numerical details and results from our training.

## F.1  Mixed Training

Comprehensive results for models on our benchmarks during mixed training are shown in Table 6 and Table 7. Results are all averaged over 10 seeds. We omit standard error values as they are all $\sigma \leq 0.1$ in each case.

## F.2  Sequential Training

Comprehensive results for models on our benchmarks during sequential training are shown in Table 9 and Table 11. Results are all averaged over 10 seeds. We report standard errors for all performance metrics.

| Model | Toy | | | Advanced | | |
|---|---|---|---|---|---|---|
| _Tools→_ | **No** | **Yes** | (↑↓) | **No** | **Yes** | (↑↓) |
| OPT-125M | 44.4 | 99.9 | ↑55.5 | 16.9 | 97.1 | ↑80.2 |
| OPT-350M | 43.3 | 100.0 | ↑56.7 | 17.8 | 97.6 | ↑79.8 |
| OPT-1.3B | 91.3 | 100.0 | ↑8.6 | 36.7 | 97.9 | ↑61.2 |
| OPT-2.7B | 95.5 | 100.0 | ↑4.5 | 41.9 | 99.3 | ↑57.4 |
| OPT-6.7B | 98.0 | 100.0 | ↑2.0 | 46.2 | 99.1 | ↑52.9 |
| OPT-13B | 99.0 | 100.0 | ↑1.0 | 50.0 | 99.4 | ↑49.4 |

Table 6: Accuracy of models on both benchmarks when data is mixed into a single task. These represent upper bounds for performance. Colored boxes refer to the increase or decrease in accuracy performance from the non-tool to tool setup, with green representing a positive change and red a negative change.

| Model | Raw Ans. | API Call | API Ans. |
|---|---|---|---|
| OPT-125M | 69.7 | 99.9 | 91.4 ↑21.57 |
| OPT-350M | 74.6 | 99.9 | 91.4 ↑16.8 |
| OPT-1.3B | 80.4 | 100.0 | 91.4 ↑11.0 |
| OPT-2.7B | 82.0 | 100.0 | 91.4 ↑9.4 |
| OPT-6.7B | 87.5 | 100.0 | 91.4 ↑3.9 |
| OPT-13B | 88.2 | 100.0 | 91.4 ↑3.2 |

Table 7: Performance of models on our continual GLUE setup when tasks are mixed. Comparison between non-tool LLMs and tool LLMs is done using the tool output.

| | Model | Without Tools | | | With Tools | | |
|---|---|---|---|---|---|---|---|
| | | Acc. | For. | L-A. | Acc. | For. | L-A. |
| **Toy Benchmark** | OPT-125M | 8.6 | 84.9 | 24.7 | 25.0 ↑16.4 | 100.0 ↑15.1 | 99.8 ↑75.1 |
| | OPT-350M | 9.0 | 92.9 | 25.7 | 25.0 ↑16.0 | 100.0 ↑7.1 | 99.8 ↑74.1 |
| | OPT-1.3B | 30.3 | 89.7 | 83.1 | 25.0 ↓5.3 | 100.0 ↑10.3 | 100.0 ↑16.9 |
| | OPT-2.7B | 31.1 | 89.9 | 89.4 | 26.0 ↓5.1 | 98.6 ↑8.8 | 100.0 ↑10.6 |
| | OPT-6.7B | 26.0 | 97.1 | 93.0 | 25.0 ↓1.0 | 100.0 ↑2.9 | 100.0 ↑7.0 |
| | OPT-13B | 28.4 | 98.1 | 95.3 | 25.0 ↓1.0 | 100.0 ↑1.9 | 100.0 ↑4.7 |
| | OPT-125M$_{+ER}$ | 36.9 | 5.8 | 26.0 | 99.9 ↑63.0 | 0.1 ↓5.7 | 99.8 ↑73.8 |
| | OPT-350M$_{+ER}$ | 33.7 | 13.2 | 27.5 | 99.9 ↑66.2 | 0.0 ↓13.2 | 99.8 ↑72.2 |
| | OPT-1.3B$_{+ER}$ | 80.8 | 8.9 | 81.8 | 100.0 ↑9.2 | 0.0 ↓2.9 | 100.0 ↑18.2 |
| | OPT-2.7B$_{+ER}$ | 83.6 | 8.7 | 89.5 | 100.0 ↑6.4 | 0.0 ↓1.7 | 100.0 ↑10.5 |
| | OPT-6.7B$_{+ER}$ | 87.9 | 12.5 | 93.8 | 100.0 ↑12.1 | 0.1 ↓12.4 | 100.0 ↑6.2 |
| | OPT-13B$_{+ER}$ | 89.8 | 12.4 | 94.9 | 100.0 ↑10.2 | 0.0 ↓12.4 | 100.0 ↑5.1 |
| **Advanced Benchmark** | OPT-125M | 10.3 | 39.3 | 15.8 | 38.4 ↑28.1 | 70.8 ↑31.5 | 95.1 ↑79.3 |
| | OPT-350M | 7.0 | 55.9 | 15.5 | 25.6 ↑18.7 | 85.8 ↑29.9 | 96.6 ↑81.1 |
| | OPT-1.3B | 25.4 | 36.9 | 35.6 | 33.4 ↑8.0 | 77.2 ↑40.3 | 98.5 ↑62.9 |
| | OPT-2.7B | 32.1 | 33.2 | 41.4 | 33.7 ↑1.6 | 76.9 ↑48.6 | 98.7 ↑57.3 |
| | OPT-6.7B | 31.6 | 40.6 | 47.3 | 32.2 ↑0.7 | 78.7 ↑38.0 | 98.9 ↑51.7 |
| | OPT-13B | 32.5 | 39.5 | 47.2 | 33.2 ↑0.7 | 78.4 ↑38.0 | 99.0 ↑51.8 |
| | OPT-125M$_{+ER}$ | 15.5 | 12.6 | 16.2 | 96.1 ↑80.6 | 1.0 ↓11.6 | 92.8 ↑76.6 |
| | OPT-350M$_{+ER}$ | 15.4 | 13.0 | 16.3 | 96.8 ↑81.4 | 0.4 ↓12.6 | 95.6 ↑79.3 |
| | OPT-1.3B$_{+ER}$ | 34.5 | 11.6 | 35.1 | 96.6 ↑62.1 | 0.5 ↓11.1 | 97.9 ↑61.8 |
| | OPT-2.7B$_{+ER}$ | 38.9 | 13.1 | 39.7 | 97.4 ↑58.5 | 0.6 ↓12.5 | 97.7 ↑57.0 |
| | OPT-6.7B$_{+ER}$ | 41.6 | 12.5 | 42.1 | 97.8 ↑56.6 | 0.5 ↓12.0 | 98.7 ↑55.6 |
| | OPT-13B$_{+ER}$ | 42.4 | 11.8 | 44.2 | 97.7 ↑56.3 | 0.5 ↓11.3 | 98.9 ↑54.7 |

Table 8: Performance on the continual learning version of our arithmetic benchmarks. Performance is noted in terms of accuracy (Acc.), forgetting (For.) and learning accuracy (L-A.), both when learning with or without tools (averaged across 10 seeds). ↑ and ↓ indicate direction of performance change when using tools, with green/red indicating if the change is desirable or undesirable. +ER indicates models use a replay buffer of 64 samples times the number of tasks.

| | Model | Without Tools | | | With Tools | | |
|---|---|---|---|---|---|---|---|
| | | Acc. (↑) | For. (↓) | L-A. (↑) | Acc. (↑) | For. (↓) | L-A. (↑) |
| **Toy Benchmark** | OPT-125M | $8.6_{\pm1.0}$ | $84.9_{\pm1.7}$ | $24.7_{\pm0.1}$ | $25.0_{\pm0.0}$ | $100.0_{\pm0.0}$ | $99.8_{\pm0.0}$ |
| | OPT-350M | $9.0_{\pm1.0}$ | $92.9_{\pm0.9}$ | $25.7_{\pm0.1}$ | $25.0_{\pm0.0}$ | $100.0_{\pm0.0}$ | $99.8_{\pm0.0}$ |
| | OPT-1.3B | $30.3_{\pm0.6}$ | $89.7_{\pm0.8}$ | $83.1_{\pm0.8}$ | $25.0_{\pm0.0}$ | $100.0_{\pm0.0}$ | $100.0_{\pm0.0}$ |
| | OPT-2.7B | $31.1_{\pm0.7}$ | $89.9_{\pm1.1}$ | $89.4_{\pm0.5}$ | $26.0_{\pm0.3}$ | $98.6_{\pm0.4}$ | $100.0_{\pm0.0}$ |
| | OPT-6.7B | $26.0_{\pm0.4}$ | $97.1_{\pm0.5}$ | $93.0_{\pm0.2}$ | $25.0_{\pm0.0}$ | $100.0_{\pm0.0}$ | $100.0_{\pm0.0}$ |
| | OPT-13B | $28.4_{\pm0.3}$ | $98.1_{\pm0.4}$ | $95.3_{\pm0.1}$ | $25.0_{\pm3.4}$ | $100.0_{\pm0.0}$ | $100.0_{\pm0.0}$ |
| | OPT-125M+ER | $36.9_{\pm0.6}$ | $5.8_{\pm1.7}$ | $26.0_{\pm0.4}$ | $99.9_{\pm0.0}$ | $0.1_{\pm0.0}$ | $99.8_{\pm0.0}$ |
| | OPT-350M+ER | $33.7_{\pm0.5}$ | $13.2_{\pm1.8}$ | $27.5_{\pm0.2}$ | $99.9_{\pm0.0}$ | $0.0_{\pm0.0}$ | $99.8_{\pm0.1}$ |
| | OPT-1.3B+ER | $80.8_{\pm0.5}$ | $8.9_{\pm0.5}$ | $81.8_{\pm2.5}$ | $100.0_{\pm0.0}$ | $0.0_{\pm0.0}$ | $100.0_{\pm0.0}$ |
| | OPT-2.7B+ER | $83.6_{\pm0.4}$ | $8.7_{\pm0.3}$ | $89.5_{\pm1.1}$ | $100.0_{\pm0.0}$ | $0.0_{\pm0.0}$ | $100.0_{\pm0.0}$ |
| | OPT-6.7B+ER | $87.9_{\pm0.8}$ | $12.5_{\pm0.9}$ | $93.8_{\pm0.5}$ | $100.0_{\pm0.0}$ | $0.1_{\pm0.0}$ | $100.0_{\pm0.0}$ |
| | OPT-13B+ER | $89.8_{\pm0.7}$ | $12.4_{\pm1.1}$ | $94.9_{\pm0.6}$ | $100.0_{\pm0.0}$ | $0.0_{\pm0.0}$ | $100.0_{\pm0.0}$ |
| **Advanced Benchmark** | OPT-125M | $10.3_{\pm0.7}$ | $39.3_{\pm7.2}$ | $15.8_{\pm0.1}$ | $38.4_{\pm3.4}$ | $70.8_{\pm4.0}$ | $95.1_{\pm0.7}$ |
| | OPT-350M | $7.0_{\pm0.7}$ | $55.9_{\pm6.2}$ | $15.5_{\pm0.3}$ | $25.6_{\pm2.4}$ | $85.8_{\pm2.9}$ | $96.6_{\pm0.2}$ |
| | OPT-1.3B | $25.4_{\pm1.7}$ | $36.9_{\pm4.8}$ | $35.6_{\pm0.3}$ | $33.4_{\pm3.9}$ | $77.2_{\pm4.6}$ | $98.5_{\pm0.0}$ |
| | OPT-2.7B | $32.1_{\pm1.9}$ | $33.2_{\pm4.2}$ | $41.4_{\pm0.4}$ | $33.7_{\pm4.2}$ | $76.9_{\pm4.9}$ | $98.7_{\pm0.0}$ |
| | OPT-6.7B | $31.6_{\pm1.8}$ | $40.6_{\pm4.2}$ | $47.3_{\pm0.2}$ | $32.2_{\pm3.0}$ | $78.7_{\pm3.5}$ | $98.9_{\pm0.0}$ |
| | OPT-13B | $32.5_{\pm1.2}$ | $39.5_{\pm3.6}$ | $47.2_{\pm0.2}$ | $33.2_{\pm2.8}$ | $78.4_{\pm3.2}$ | $99.0_{\pm0.0}$ |
| | OPT-125M+ER | $15.5_{\pm0.3}$ | $12.6_{\pm2.3}$ | $16.2_{\pm0.1}$ | $96.1_{\pm0.2}$ | $1.0_{\pm0.1}$ | $92.8_{\pm0.8}$ |
| | OPT-350M+ER | $15.4_{\pm0.1}$ | $13.0_{\pm1.4}$ | $16.3_{\pm0.2}$ | $96.8_{\pm0.1}$ | $0.4_{\pm0.1}$ | $95.6_{\pm0.3}$ |
| | OPT-1.3B+ER | $34.5_{\pm0.2}$ | $11.6_{\pm1.9}$ | $35.1_{\pm0.2}$ | $96.6_{\pm0.2}$ | $0.5_{\pm0.1}$ | $97.9_{\pm0.9}$ |
| | OPT-2.7B+ER | $38.9_{\pm0.1}$ | $13.1_{\pm1.1}$ | $39.7_{\pm0.1}$ | $97.4_{\pm0.3}$ | $0.6_{\pm0.1}$ | $97.7_{\pm0.5}$ |
| | OPT-6.7B+ER | $41.6_{\pm0.1}$ | $12.5_{\pm1.4}$ | $42.1_{\pm0.1}$ | $97.8_{\pm0.2}$ | $0.5_{\pm0.1}$ | $98.7_{\pm0.8}$ |
| | OPT-13B+ER | $42.4_{\pm0.1}$ | $11.8_{\pm1.2}$ | $44.2_{\pm0.1}$ | $97.7_{\pm0.2}$ | $0.5_{\pm0.1}$ | $98.9_{\pm0.7}$ |

Table 9: Performance of models on the continual learning version of our arithemtic benchmarks. Given that each tool in these settings are oracles, API call correctness and final answer correctness with tools is the same. Performance is noted in terms of accuracy (Acc.), forgetting (For.) and learning accuracy (L-A.), both when learning with or without tools (averaged across 10 seeds). ↑ indicates that higher values are better, ↓ lower. +ER indicates models use a replay buffer of 64 samples times the number of tasks.

| Model | Without Tools | | | With Tools | | |
|---|---|---|---|---|---|---|
| | Acc. | For. | L-A. | Acc. | For. | L-A. |
| OPT-125M | 27.0 | 92.4 | 56.5 | 23.4 ↓6.6 | 100.0 ↑8.6 | 91.4 ↑34.9 |
| OPT-350B | 27.2 | 94.4 | 57.3 | 23.3 ↓6.9 | 100.0 ↑5.6 | 91.5 ↑34.2 |
| OPT-1.3B | 27.2 | 89.7 | 67.3 | 23.4 ↓3.8 | 99.6 ↑9.9 | 91.4 ↑24.1 |
| OPT-2.7B | 28.9 | 93.2 | 69.2 | 23.5 ↓5.4 | 99.4 ↑6.2 | 91.2 ↑22.0 |
| OPT-6.7B | 30.2 | 92.7 | 71.7 | 23.6 ↓6.6 | 100.0 ↑7.3 | 91.4 ↑19.7 |
| OPT-13B | 29.9 | 93.4 | 72.1 | 23.5 ↓6.4 | 100.0 ↑6.6 | 91.4 ↑19.3 |
| OPT-125M$_{+ER}$ | 60.7 | 1.9 | 62.2 | 91.3 ↑30.6 | 0.1 ↓1.8 | 91.4 ↑29.2 |
| OPT-350M$_{+ER}$ | 63.4 | 1.8 | 66.2 | 91.3 ↑27.9 | 0.1 ↓1.8 | 91.4 ↑25.2 |
| OPT-1.3B$_{+ER}$ | 68.4 | 2.1 | 70.3 | 91.3 ↑22.9 | 0.1 ↓2.0 | 91.4 ↑21.1 |
| OPT-2.7B$_{+ER}$ | 68.6 | 2.1 | 70.4 | 91.4 ↑22.8 | 0.1 ↓2.0 | 91.5 ↑21.1 |
| OPT-6.7B$_{+ER}$ | 71.7 | 2.0 | 73.4 | 91.4 ↑19.7 | 0.0 ↓2.0 | 91.4 ↑18.0 |
| OPT-13B$_{+ER}$ | 71.5 | 2.2 | 73.7 | 91.4 ↑19.9 | 0.1 ↓2.1 | 91.4 ↑17.7 |

Table 10: Performance of models on our continual GLUE setup (10 seeds). Performance gain is compared using correctness of the final tool response rather than the API call correctness. Notation is the same as in Table 8.

| Model | Without Tools | | | With Tools (API Call) | | | With Tools (Final Answer) | | |
|---|---|---|---|---|---|---|---|---|---|
| | Acc. (↑) | For. (↓) | L-A. (↑) | Acc. (↑) | For. (↓) | L-A. (↑) | Acc. (↑) | For. (↓) | L-A. (↑) |
| OPT-125M | $27.0_{\pm1.1}$ | $92.4_{\pm4.6}$ | $56.5_{\pm2.9}$ | $25.4_{\pm1.5}$ | $100.0_{\pm0.0}$ | $99.3_{\pm0.0}$ | $23.4_{\pm1.4}$ | $100.0_{\pm0.0}$ | $91.4_{\pm0.1}$ |
| OPT-350B | $27.2_{\pm0.9}$ | $94.4_{\pm5.3}$ | $57.3_{\pm2.9}$ | $25.3_{\pm1.6}$ | $100.0_{\pm0.0}$ | $99.2_{\pm0.0}$ | $23.3_{\pm1.4}$ | $100.0_{\pm0.0}$ | $91.5_{\pm0.0}$ |
| OPT-1.3B | $27.2_{\pm1.4}$ | $89.7_{\pm5.3}$ | $67.3_{\pm2.9}$ | $25.4_{\pm1.3}$ | $99.9_{\pm0.0}$ | $99.2_{\pm0.1}$ | $23.4_{\pm1.1}$ | $99.7_{\pm0.1}$ | $91.4_{\pm0.0}$ |
| OPT-2.7B | $28.9_{\pm1.8}$ | $93.2_{\pm3.9}$ | $69.2_{\pm3.6}$ | $25.6_{\pm1.6}$ | $99.9_{\pm0.1}$ | $99.0_{\pm0.0}$ | $23.5_{\pm1.4}$ | $99.4_{\pm0.3}$ | $91.2_{\pm0.1}$ |
| OPT-6.7B | $30.2_{\pm1.2}$ | $92.7_{\pm4.3}$ | $71.7_{\pm3.5}$ | $25.8_{\pm1.4}$ | $99.9_{\pm0.1}$ | $99.3_{\pm0.0}$ | $23.6_{\pm1.2}$ | $100.0_{\pm0.0}$ | $91.4_{\pm0.1}$ |
| OPT-13B | $29.9_{\pm1.1}$ | $93.4_{\pm3.4}$ | $72.1_{\pm3.2}$ | $25.7_{\pm1.4}$ | $99.9_{\pm0.1}$ | $99.2_{\pm0.0}$ | $23.5_{\pm1.2}$ | $100.0_{\pm0.0}$ | $91.4_{\pm0.1}$ |
| OPT-125M$_{+ER}$ | $60.7_{\pm0.6}$ | $1.9_{\pm0.5}$ | $62.2_{\pm0.8}$ | $99.2_{\pm0.1}$ | $0.1_{\pm0.1}$ | $99.3_{\pm0.1}$ | $91.3_{\pm0.1}$ | $0.1_{\pm0.1}$ | $91.4_{\pm0.1}$ |
| OPT-350M$_{+ER}$ | $63.4_{\pm0.5}$ | $1.8_{\pm0.4}$ | $66.2_{\pm0.8}$ | $99.3_{\pm0.1}$ | $0.1_{\pm0.1}$ | $99.4_{\pm0.1}$ | $91.3_{\pm0.1}$ | $0.1_{\pm0.1}$ | $91.4_{\pm0.1}$ |
| OPT-1.3B$_{+ER}$ | $68.4_{\pm0.7}$ | $2.1_{\pm0.6}$ | $70.3_{\pm0.9}$ | $99.2_{\pm0.1}$ | $0.1_{\pm0.1}$ | $99.3_{\pm0.1}$ | $91.3_{\pm0.1}$ | $0.1_{\pm0.1}$ | $91.4_{\pm0.1}$ |
| OPT-2.7B$_{+ER}$ | $68.6_{\pm0.6}$ | $2.1_{\pm0.5}$ | $70.4_{\pm1.1}$ | $99.4_{\pm0.1}$ | $0.0_{\pm0.1}$ | $99.5_{\pm0.1}$ | $91.4_{\pm0.1}$ | $0.1_{\pm0.1}$ | $91.5_{\pm0.1}$ |
| OPT-6.7B$_{+ER}$ | $71.7_{\pm0.8}$ | $2.0_{\pm0.6}$ | $73.4_{\pm0.9}$ | $99.4_{\pm0.1}$ | $0.0_{\pm0.1}$ | $99.4_{\pm0.1}$ | $91.4_{\pm0.1}$ | $0.0_{\pm0.1}$ | $91.4_{\pm0.1}$ |
| OPT-13B$_{+ER}$ | $71.5_{\pm0.7}$ | $2.2_{\pm0.4}$ | $73.7_{\pm1.0}$ | $99.4_{\pm0.1}$ | $0.1_{\pm0.1}$ | $99.5_{\pm0.1}$ | $91.4_{\pm0.1}$ | $0.1_{\pm0.1}$ | $91.4_{\pm0.1}$ |

Table 11: Average accuracy, forgetting and learning accuracy for models on our continual GLUE setup (averaged across 10 seeds). Tool performance is separated into the ability to generate the correct API call as well as the ability of the tool to produce the correct response given the correct API call (incorrect API calls are treated as incorrect).