# OpenReview forum: "Towards Practical Tool Usage for Continually Learning Large Language Models"
_TMLR — Rejected by TMLR_

### Review · Reviewer_zXRK · 2024-09-07

**Summary Of Contributions:**

- This research studies an interesting and novel question. This work investigates the impact of tool usage, as opposed to relying solely on parametric knowledge, within the context of continual learning. The motivation behind this study lies in the hypothesis that learning to utilize tools is inherently more generalizable and adaptable compared to memorizing answers, a problem that remains relatively underexplored in the current literature.

- The paper introduces an innovative experimental setup by applying continual learning techniques, such as replay buffers, both with and without the use of tools, to measure model's forgetting patterns.

**Audience:**

Yes

**Claims And Evidence:**

No

**Requested Changes:**

- Discuss and explain the key differences with Leo et al., 2023.
- Design more challenging experiments so that the tool use case is not artificially too easy.
- Use more recent open-weight models such as LLama2, LLama3 instead of OPT. Although the choice of the underlying model is orthogonal to the experiments, model behaviors may alter drastically when they are sufficiently trained.

**Strengths And Weaknesses:**

**Strengths**
- The research question proposed in this work is interesting. It explores the effect of tool usage in continual learning paradigm. This is a practical problem.
- The experimental results demonstrate the potential of using tools to mitigate the forgetting problem. The final experiment highlights that when common continual learning techniques, such as replay buffers, are applied, models trained using tool-based data exhibit a lower forgetting rate compared to models relying solely on direct answer prediction. This finding underscores the positive impact of tool usage in constrained domains (i.e., with fewer than 10 tools).

**Weaknesses**
- Some of the research questions explored have been previously studied in the literature. The first key experiment investigates whether increasing model size enhances continual learning capabilities, concluding that even larger models face difficulties with task changes. However, this question has been thoroughly examined in previous work, such as [1], with similar findings.
- The experimental design may not sufficiently support the conclusions. A key concern is the limited number of tools considered, which creates a scenario that naturally favors tool usage over direct answer prediction.
    - With a small toolset, models only need to implicitly classify the task, select the correct tool, and fill arguments through copying or minimal rephrasing. In contrast, without tools, models must engage in more complex reasoning. The transition between tasks is more drastic in non-tool use setting. The comparison would be more compelling with a larger toolset (e.g., thousands of tools with complex argument configurations), chaining of tools, where selecting and applying the correct tool becomes challenging.
    - Additionally, incorporating techniques like chain-of-thought prompting in non-tool-based training could provide a more balanced comparison.
    - Although the work additionally experiment with GLUE, each task is reduced to one tool (i.e. the classification problem is still easy), despite filling the correct arguments are slight more challenging.

Reference:

[1] Luo et al., 2023, *An Empirical Study of Catastrophic Forgetting in Large Language Models During Continual Fine-tuning*.

---

> ### Author Response · Authors · 2024-10-08
> **Author Response (1/N)**
>
> We thank the reviewer for their valuable review and feedback. We are grateful that they, like reviewer GJtr and RACG, mention the interesting and novel question being studied, as well as the innovative experimental setup. We hope that the following response that we provide can clarify some remaining questions and provide them answers that will address their comments.
>
> -----
>
> > Some of the research questions explored have been previously studied in the literature.
>
> We thank the reviewer for their comment but would like to clarify the differences between our work and that within existing literature. First, we would like to state that the primary goal of our work is to study tool usage within a continual learning framework with LLMs; there is currently no work in the existing literature that explores this specific problem. As all the reviewers agree, this is an interesting problem that is novel and has yet to be explored before us or in works that follow. While some previous works look at problems which we partially look at, namely scaling, this is not the main focus of our paper and is conducted for different purposes.
>
> To elaborate, though it is certainly true that we do provide an analysis of changes that occur with scale which has been previously discussed in existing literature, we would like to point out that the primary observations of comparing against scale are not whether or not scale itself is sufficient as a remedy to forgetting in continual learning (which is a question that is not unique to this topic but to the broader research question of continual learning as a whole). Instead, it relates to the use of tools, with scale only serving as an axis of variation that is meant as a comparison of how tools can still be effective across scale. As far as we are aware, this is the only work that explores such a setting. While there might be two questions that are not unique in the exact topic of study (continual learning with LLMs and tool-learning with LLMs), this remains the only work that explicitly makes an attempt to connect the two.
>
> > Discuss and explain the key differences with Luo et al., 2023.
>
> We are glad to provide this distinction. To clarify, Luo et al. (2023) concern themselves with the question of investigating forgetting when continually parametric fine-tuning LLMs. Unlike them, we explore this from the angle of tool usage and show that they serve as a potential way to render the problem much easier and can lead as a solution to this problem.
>
> More specially, our interest is not in explicitly investigating forgetting, but instead into whether or not learning to use tools can in fact be beneficial for continual learning. For example, they provide an evaluation of models along scale, as well as compare between base pre-trained and instruction fine-tuned models. While we do have a small set of results that have some overlap (namely analysis along the dimension of scale), these serve different purposes in our work. For Luo et al., these results are meant to demonstrate that scale cannot solve continual learning. For us, we first make the same observation, but we then make a comparison with these results between tool-learning LLMs and vanilla LLMs to show that vanilla LLMs
>
> 1. Remain more robust by being able to learn incoming tasks more effectively.
> 2. Have a more consistent behavior across model size when it comes to continual learning.
>
> Additionally, to return to our observations regarding scale, while we make this observation in Section 6.2, the goal and conclusions we reach are slightly different from the results presented by Luo et al. (2023), which is the closest comparison in existing literature (we elaborate further on this in the next point). While Luo et al. (2023) observe increasing forgetting in their experiments (in Section 5.3 of their work), as we do in some of ours, their analysis ends there. Meanwhile, we interest ourselves additionally on other trends in behavior (elaborated upon in the following point) which we attempt to understand to determine some of the potential advantages of learning tools for continual learning as opposed to standard fine-tuning as done by Luo et al. (2023).
>
> As such, while there is some overlap in the nature of the experiments and results that are presented, the end goal is quite different and we contend that we provide a more extensive analysis (grounded in the research question we ask) given the nature of the work.

---

> > ### Author Response · Authors · 2024-10-08
> > **Author Response (2/N)**
> >
> > > The experimental design may not sufficiently support the conclusions.
> >
> > We appreciate the reviewer for expressing their opinion on this matter, however we would like to further make the case that the tools and experiments we use are sufficient for grounding the claims we make. In particular, our claim through our experimental observations is that tool-learning LLMs have the _*potential*_ to serve as better continual learners. While it is certainly true that our experiments are not exhaustive, nor do they cover the entire set of ways in which tools can be used, our results and the analysis that follow in section 6 explain in detail how and why we reach these conclusions.
> >
> > We would also like to elaborate on why our design supports our conclusions. First, despite the use of tools being similar to extracting information from a context and learning how to properly in-fill a pre-specified template, this is (in its most raw form) inherently how tools are used even by humans. However, when learning individual tools in a continual manner, there nonetheless exists a distribution shift that exists, namely in how this parsing must be done and the rules of how the tool should be used. It is certainly true that this shift may not appear as strong as that faced by the non-tool learning LLMs, which must instead learn the underlying (either grammatical or logical) rules that govern the task; yet this is precisely what we are attempting to test with our setup. We demonstrate that learning this tool-based shift is significantly easier to perform compared to the underlying task shift and hence LLMs might become better suited for such settings if they learn to use tools.
> >
> > > Design more challenging experiments so that the tool use case is not artificially too easy.
> >
> > We agree that exploring more tools and more complex ways of tool use would be beneficial. However, as we mention above, we believe our conclusions and statements are qualified in such a manner that the selected tools are sufficient. However a more extensive study would be certainly useful, but some limitations with respect to being able to generate examples make it so that we believe this would be more suitable as following work.
> >
> > We also provide results where we mix all tasks across all settings together. In particular, we mix all the subtasks from each setting, then randomly select an order to learn everything continually.
> >
> > | Model | Accuracy (W/O Tools) | Forgetting (W/O Tools) | Learning Accuracy (W/O Tools) | Accuracy (W/ Tools) | Forgetting (W/ Tools) | Learning Accuracy (W/ Tools) |
> > | ------- |----------- | ------- | ------------- |----------- | ------- | ------------- |
> > | OPT-125M | 11.2 | 68.8 | 31.2 | 23.6 | 91.2 | 95.4 |
> > | OPT-350M | 12.7 | 71.2 | 35.2 | 25.1 | 93.7 | 95.8 |
> > | OPT-1.3B | 24.3 | 64.6 | 61.4 | 27.1 | 93.6 | 95.7 |
> > | OPT-2.7B | 22.2 | 62.8 | 63.2 | 26.2 | 92.4 | 96.1 |
> > | OPT-6.7B | 25.2 | 72.5 | 65.8 | 24.8 | 92.6 | 96.3 |
> > | OPT-13B | 24.5 | 64.8 | 67.4 | 26.2 | 92.1 | 96.6 |
> > | LLaMA2-7B | 24.3 | 85.8 | 76.4 | 24.2 | 94.6 | 97.2 |
> > | LLaMA2-13B | 25.4 | 83.3 | 75.3 | 24.8 | 91.3 | 98.1 |
> > | LLaMA3-8B | 25.6 | 84.4 | 75.3 | 24.5 | 93.5 | 97.7 |
> > | OPT-125M (+ER) | 38.3 | 10.2 | 43.1 | 93.2 | 1.1 | 96.1 |
> > | OPT-350M (+ER) | 37.8 | 9.8 | 41.7 | 93.1 | 0.9 | 97.3 |
> > | OPT-1.3B (+ER) | 59.7 | 7.8 | 61.3 | 94.5 | 0.7 | 97.5 |
> > | OPT-2.7B (+ER) | 65.3 | 9.5 | 71.5 | 95.5 | 0.6 | 97.5 |
> > | OPT-6.7B (+ER) | 69.6 | 8.3 | 74.4 | 96.2 | 0.4 | 98.1 |
> > | OPT-13B (+ER) | 69.4 | 10.8 | 75.2 | 96.5 | 0.3 | 98.3 |
> > | LLaMA2-7B (+ER) | 68.4 | 6.5 | 75.6 | 96.3 | 0.1 | 97.2 |
> > | LLaMA2-13B (+ER) | 69.6 | 5.1 | 73.7 | 96.6 | 0.5 | 98.6 |
> > | LLaMA3-8B (+ER) | 68.2 | 6.4 | 76.1 | 96.8 | 0.2 | 98.3 |
> >
> > These results follow our observations from using a single setting, showing that the same trends hold here. This serves as further justification regarding how the nature of the tasks is more or less independent from the difficulty of learning.
> >
> > Additionally, using LLMs with tools is a unique scenario as their core learning principle is template matching; the complexity comes from the parsing the right arguments for the learnt tool template. We are supportive of the claim that additionally experimenting with increasingly diverse tool-sets is necessary to produce a fool-proof claim regarding how tool-learning might be able to help LLMs learn continually. However, as mentioned by Reviewer GJtr, designing these settings can be quite complicated and requires an immense amount of resources. As such, our claims in the paper again does not make the direct jump of claiming that tool learning LLMs can solve continual learning problems, but rather that they show the potential to be continual learners in comparison to standard LLMs. We believe such a claim is supported by the evidence from the experiments we have conducted and believe that these suggested challenging experiments can serve as a next step towards making even stronger claims regarding tool learning LLMs and their potential.

---

> ### Author Response · Authors · 2024-10-08
> **Author Response (3/N)**
>
> > Use more recent open-weight models such as LLama2, LLama3 instead of OPT.
>
> We provide results on LLaMA models that we can feasibly train. Our below results show that our observations with OPT persist in LLaMA models, with the use of a small replay buffer being sufficient to significantly boost continual learning performance while directly fine-tuning on the tasks consistently results in lower performance compared to direct tool usage.
>
> - Toy Arithmetic
>
> | Model | Accuracy (W/O Tools) | Forgetting (W/O Tools) | Learning Accuracy (W/O Tools) | Accuracy (W/ Tools) | Forgetting (W/ Tools) | Learning Accuracy (W/ Tools) |
> | ------- |----------- | ------- | ------------- |----------- | ------- | ------------- |
> | LLaMA2-7B | 25.4 | 98.4 | 95.6 | 25.0 | 100.0 | 100.0|
> | LLaMA2-13B | 25.7 | 98.9 | 96.2 | 25.0| 100.0 | 100.0 |
> | LLaMA3-8B | 25.6 | 99.3 | 96.7 | 25.0 | 100.0 | 100.0|
> | LLaMA2-7B (+ER) | 93.2 | 2.4 | 96.0 | 99.9 | 0.1 | 100.0|
> | LLaMA2-13B (+ER) | 94.1 | 2.3 | 96.3 | 100.0| 0.0 | 100.0 |
> | LLaMA3-8B (+ER) | 93.4 | 1.9 | 96.6 | 100.0 | 0.0 | 100.0|
>
> - Advanced Arithmetic
>
> | Model | Accuracy (W/O Tools) | Forgetting (W/O Tools) | Learning Accuracy (W/O Tools) | Accuracy (W/ Tools) | Forgetting (W/ Tools) | Learning Accuracy (W/ Tools) |
> | ------- |----------- | ------- | ------------- |----------- | ------- | ------------- |
> | LLaMA2-7B | 17.3 | 64.8 | 63.4 | 34.3 | 75.4 | 99.1 |
> | LLaMA2-13B | 17.9 | 64.4 | 64.3 | 33.8 | 78.3 | 99.2 |
> | LLaMA3-8B | 17.2 | 65.1 | 63.3 | 34.1 | 76.3 | 99.0 |
> | LLaMA2-7B (+ER) | 58.6 | 6.5 | 63.5 | 98.4 | 0.5 | 98.9 |
> | LLaMA2-13B (+ER) | 59.1 | 7.1 | 63.9 | 98.5 | 0.7 | 99.1 |
> | LLaMA3-8B (+ER) | 58.2 | 6.4 | 63.6 | 98.6 | 0.6 | 99.2 |
>
> - GLUE
>
> | Model | Accuracy (W/O Tools) | Forgetting (W/O Tools) | Learning Accuracy (W/O Tools) | Accuracy (W/ Tools) | Forgetting (W/ Tools) | Learning Accuracy (W/ Tools) |
> | ------- |----------- | ------- | ------------- |----------- | ------- | ------------- |
> | LLaMA2-7B | 28.4 |96.7| 80.7 | 23.4 | 100.0 | 91.3 |
> | LLaMA2-13B | 28.6 |97.4| 81.2 | 23.4 | 100.0 | 91.4 |
> | LLaMA3-8B | 28.5 |96.5| 80.6 | 23.4 | 100.0 | 91.4 |
> | LLaMA2-7B (+ER) | 80.1 |3.8| 82.8 | 91.4 | 0.0 | 91.4|
> | LLaMA2-13B (+ER) | 81.2 |4.2| 84.3 | 91.3 | 0.1 | 91.4|
> | LLaMA3-8B (+ER) | 80.3 |4.1| 83.2 | 91.4 | 0.0 | 91.4|
>
> -----
>
> Again, we appreciate the reviewer’s careful comments and hope that the provided response is sufficient to address their remaining questions and/or concerns while providing the requested changes.

---

### Review · Reviewer_GJtr · 2024-09-09

**Summary Of Contributions:**

The authors shows that, in a continual learning setting (one task after another), model learning to completing tasks by calling tools is more beneficial than learning how to solve the tasks by the model itself.
The series of tasks are math tasks and natural language inference/classification tasks. They also shows that using data replay really helps with the continual setting.

**Audience:**

Yes

**Broader Impact Concerns:**

No ethical implication, since the tasks are pretty generic.

**Claims And Evidence:**

No

**Requested Changes:**

See some related comments in Weakness.

* Include and report baseline “Mixed Dataset”.
* Conduct experiments to control the imperfect-ness of tools. And motivate a continual learning setting that illustrates the importance of dealing with incomplete tools. [1] gives a good example of complex task and imperfect tool.
* Could the author provide examplary input and output to the model in tool and non-tool scenario?
* Does simple prompting LM to use tools works? And what is model's performance without any training (for math and glue)? Could author report them as baselines? Have the author tried different decoding settings/hypers.
* Could the experiments include more complex tool-use setting?
  * The settings are too simplistic -- the model probably only need prompting to do well and the capability tested are too primitive --- that I don't see a need to do continual learning. Or the continual learning results here are not extrapolatable to a "real" setting. Even with a math setting, the task complexity needs to be more than what's currently are.

[1] LLMs in the Imaginarium: Tool Learning through Simulated Trial and Error

**Strengths And Weaknesses:**

# Strength:

* The framing of tool use in a continual learning setting is intriguing.
* It’s interesting to know that replay makes a significant difference in tool setting.

# Weakness:

* **The experiment are not solid** and I am feeling the analysis in section 6 (and appendix) are based on the same set of experiments. See requested changes.
  * Section 6.2 talks about results on “More parameters does not negate forgetting”, but Fig 4 is just show a subset of Figure 2b+c. Same for 6.3; figure 5 shows a subset of Fig 2b; same for Figure 3.
  * Appendix in table are just quantitative values for Figure 2. But I think showing this is valid.
  * Therefore, the length of the paper could essentially be shrinked significantly.
* Missing results/figures. At the end of Section 6.1, the authors mentioned “But, contrary to Figure 3…”; however, no evidence is included in the paper.
* For the discussion section (sec 7), I don’t see a big conceptual difference from related work section (sec 2).
* I don not follows Why Q3 (“How do tool-augmented LLMs fare with imperfect tools?”) follows from the main question “Can learning to use tools alleviate sequential learning challenges?”, since Q3 stands on its own and not really requires a continual setting.
* The design of the experiments are flawed.
  * The experiment uses basic arithmetic operation as the tool use. However, LLM are just not designed/expected to perform numerical operation [1]. Stuff like writing code or Lean-style proof could be a middle ground since the output sequences remains at a logic level. Due to lack of experience, I do not know what tools could be used here. Therefore, I do not agree that things hap
  * The authors did not describe how the language model gives final answer with imperfect tool.

> Explicit rules exist for arithmetic tasks; if one designs a tool to follow said rules, then mastering the tool is equivalent to mastering the task.

The authors talked about "rules" in such a generic sense, but the rules in math and rules in language understanding are so different, and not necessarily comparable so I don't think whatever is found in one setting could be extrapolated to another setting, not to mention even more realistic setting.

The authors include NLI tasks; however, it's unclear why LLM is expected to use tools for this, since LM could already do quite well for this. In Toolformer paper, the tools includes QA and Machine Translation, which I think the task is complex enough that a dedicated tool is useful. However, I do not this is the case for NLI tasks included here.

[1] Faith and Fate: Limits of Transformers on Compositionality

---

> ### Author Response · Authors · 2024-10-08
> **Author Response (1/N)**
>
> We thank the reviewer for their valuable review and feedback. We are grateful that like the other reviewers, they find the problem and results interesting, with the results showing some potential interesting directions. We hope the following response can address some of their concerns.
>
> -----
>
> > Therefore, the length of the paper could essentially be shrinked significantly.
>
> It is true that some of the results we present in our analysis (section 6) are a subset of the results from Figure 2. The goal here was to make use of permitted space to make it easier to find the exact results and render things more visible. We are happy to reduce the number of plots if the reviewer believes that is beneficial for readability.
>
> > Missing results/figures. At the end of Section 6.1, the authors mentioned “But, contrary to Figure 3…”; however, no evidence is included in the paper.
>
> The claim being made here is that even if we mix all the data together and fine-tuning (thereby rendering the task no longer a continual learning process), there still remains a gap that exists between the performance with tools and that without. This sentence is meant to indicate that Figure 3 demonstrates this gap to exist (see the gray bar above the red bar, which is the difference between performance with tools and without), indicating that tool learning can still be beneficial.
>
> However, the reviewer was likely confused by the previous sentence, which we acknowledge included a minor error that was not caught. We will qualify this by instead stating “*The hypothesis is that if the LMs show significant retention as indicated with the comparable performances to using tools, then performance will be nearly equivalent in both cases as each model will be able to correctly learn the associated logic necessary for the task*”, which we believe is a more apt adjustment that is supported by our results.
>
> > For the discussion section (sec 7), I don’t see a big conceptual difference from the related work section (sec 2).
>
> We are willing to merge aspects of these sections. The related work is meant to provide a background of the problem to the reader, whereas the discussion section is a more grounded discussion based on the results we observe.
>
> > I do not follow why Q3 (“How do tool-augmented LLMs fare with imperfect tools?”) follows from the main question “Can learning to use tools alleviate sequential learning challenges?”, since Q3 stands on its own and (does) not really require a continual setting.
>
> We acknowledge that perhaps the phrasing could be somewhat better on this front, but we would like to justify why Q3 (“How do tool-augmented LLMs fare with imperfect tools?”) is an interesting question and why it requires a continual learning setting. As is explicitly mentioned, we wanted to compare the effects of using tools for continual learning when the tools are imperfect, which is when the tool itself can be a source of error even if the call is correct. This is interesting because there are two places where errors can be introduced: 1) generating the tool call and 2) using the tool. Compared to generating directly, where there is only one explicit area where the error is induced, we wanted to compare what would occur when using this type of tool.
>
> Furthermore, part of this investigation relates to the base model capacity and that of the tool model. In particular, given a much larger model that can generally learn adequately well on the task and a smaller tool model that is not perfect, does there eventually exist a point at which the use of the model learning on the task directly outperform the tool? What we observe here is that even with a 13 billion parameter OPT model that when fine-tuned directly for classification can outperform the tool model, by switching the task to generation the viability of using the model for continual learning decreases significantly within this setting and tools remain a very useful alternative in this scenario, despite the imperfect nature of the tool (ie. the fact that it cannot achieve perfect accuracy on the task, unlike the other tasks where the tool is trivially suited for the task.)

---

> > ### Author Response · Authors · 2024-10-08
> > **Author Response (2/N)**
> >
> > > The authors talked about "rules" in such a generic sense, but the rules in math and rules in language understanding are so different, and not necessarily comparable so I don't think whatever is found in one setting could be extrapolated to another setting, not to mention even more realistic setting(s).
> >
> > We do appreciate the reviewer for expressing their concerns. Nevertheless, we would like to elaborate on why our design supports our conclusions. The goal of continual learning is to find methods to overcome the shifts in distribution that exist between the tasks being learned. Evidently, in the non-tool learning setup, this requires learning some abstract set of rules and attempting to retain them as the tasks change, making it likely increasingly difficult as the number of tasks increase. While the reviewer has concerns that using tools simplifies this problem, this is exactly the problem we are attempting to address.
> >
> > Firstly, it is certainly possible and we are also inclined to believe that tools will simplify this by offloading the complexity and rule-based processes to the tool itself. This is why we are motivated to study this specific setting, as it is of interest to know if by removing this complexity, the LLM might have a greater proficiency for continually learning tasks. However, when learning individual tools in a continual manner, a distribution shift nonetheless exists, namely in how this parsing must be done and the rules of how the tool should be used. While this shift may not appear as strong as that faced by the non-tool learning LLMs, this is precisely what we are attempting to test with our setup. We demonstrate that learning this tool-based shift is significantly easier to perform compared to the underlying task shift and hence LLMs might become better suited for such settings if they learn to use tools.
> >
> > As such, despite the synthetic nature of the tasks, we believe the setup itself is valid and enables us to validate the conclusions we make based on our results. To supplement this, we provide results from an additional experiment where we combine all the (sub)tasks from each of our settings and learn them within a larger continual learning setup.
> >
> > | Model | Accuracy (W/O Tools) | Forgetting (W/O Tools) | Learning Accuracy (W/O Tools) | Accuracy (W/ Tools) | Forgetting (W/ Tools) | Learning Accuracy (W/ Tools) |
> > | ------- |----------- | ------- | ------------- |----------- | ------- | ------------- |
> > | OPT-125M | 11.2 | 68.8 | 31.2 | 23.6 | 91.2 | 95.4 |
> > | OPT-350M | 12.7 | 71.2 | 35.2 | 25.1 | 93.7 | 95.8 |
> > | OPT-1.3B | 24.3 | 64.6 | 61.4 | 27.1 | 93.6 | 95.7 |
> > | OPT-2.7B | 22.2 | 62.8 | 63.2 | 26.2 | 92.4 | 96.1 |
> > | OPT-6.7B | 25.2 | 72.5 | 65.8 | 24.8 | 92.6 | 96.3 |
> > | OPT-13B | 24.5 | 64.8 | 67.4 | 26.2 | 92.1 | 96.6 |
> > | LLaMA2-7B | 24.3 | 85.8 | 76.4 | 24.2 | 94.6 | 97.2 |
> > | LLaMA2-13B | 25.4 | 83.3 | 75.3 | 24.8 | 91.3 | 98.1 |
> > | LLaMA3-8B | 25.6 | 84.4 | 75.3 | 24.5 | 93.5 | 97.7 |
> > | OPT-125M (+ER) | 38.3 | 10.2 | 43.1 | 93.2 | 1.1 | 96.1 |
> > | OPT-350M (+ER) | 37.8 | 9.8 | 41.7 | 93.1 | 0.9 | 97.3 |
> > | OPT-1.3B (+ER) | 59.7 | 7.8 | 61.3 | 94.5 | 0.7 | 97.5 |
> > | OPT-2.7B (+ER) | 65.3 | 9.5 | 71.5 | 95.5 | 0.6 | 97.5 |
> > | OPT-6.7B (+ER) | 69.6 | 8.3 | 74.4 | 96.2 | 0.4 | 98.1 |
> > | OPT-13B (+ER) | 69.4 | 10.8 | 75.2 | 96.5 | 0.3 | 98.3 |
> > | LLaMA2-7B (+ER) | 68.4 | 6.5 | 75.6 | 96.3 | 0.1 | 97.2 |
> > | LLaMA2-13B (+ER) | 69.6 | 5.1 | 73.7 | 96.6 | 0.5 | 98.6 |
> > | LLaMA3-8B (+ER) | 68.2 | 6.4 | 76.1 | 96.8 | 0.2 | 98.3 |
> >
> > These results follow our observations from using a single setting, showing that the same trends hold here. This serves as further justification regarding how the nature of the tasks is more or less independent from the difficulty of learning. We believe that this includes a type of scenario where the shift might be more reasonable and in-line with what the reviewer believes to be a relevant scenario from which conclusions can more appropriately be drawn.

---

> ### Author Response · Authors · 2024-10-08
> **Author Response (3/N)**
>
> > The design of the experiments are flawed.
>
> We would like to kindly counter by stating that our experiments do show, our experimental design is a well grounded combination of two rather well explored setups. First, we use a standard incremental task learning setup that is well adopted in continual learning literature. For each task, we are learning API calls, as is standard for tool-learning LLMs. We also argue that our task themselves do properly test for the properties we are looking for within these LLMs, as we motivate in each part of Section 5.
>
> We would also like to further state that while our tasks are synthetic, they do resemble real-world scenarios in which LLMs may be used. For example, we can imagine a user continually asking an LLM to summarize text and then asking it to further translate it. Then they may repeat the same process for another passage of text. This is similar to simply learning to use one tool and then moving on to the next. We admit that these tools aren’t chosen to solve what may be typically considered exciting problems for which LLMs are commonly applied in the real world, they are relevant enough in the sense that they can reflect how LLMs may be used in the real world.
>
> > Include and report baseline “Mixed Dataset”.
>
> We have provided these explicit numbers in the appendix but will also add them to all relevant figures.
>
> > Could the author provide exemplary input and output to the model in the tool and non-tool scenario(s)?
>
> We thank the reviewer for suggesting that we have perhaps not provided enough details on this front. We follow the template used by Schick et al. (2023) for tool outputs. The non-tool outputs are simply the raw answers that must be predicted. As an example for an arithmetic example
>
> Input: What is the sum of 10 and 5?
> Non-tool Output: The sum of 10 and 5 is 15.
> Tool-Output: The sum of 10 and 5 is <API> Add(10, 5) → 15 </API>.
>
> In the tool-output, the final answer is generated by executing the API call.
>
> > Does simple prompting (of the) LM to use tools work? And what is the model's performance without any training (for math and glue)? Could the authors report them as baselines?
>
> We have attempted (few-shot) prompting of the LLM (for OPT-130M to 2.7B) and observed no tangible changes, hence these results were not explicitly included. Namely, we would fine-tune the model then generate with few-shot examples provided with the test example.
>
> Without any training, performance is dependent on the model size but is quite low (< 20%) even for the largest model and the simpler arithmetic task we designed. Results are as follows on base models without fine-tuning.
>
> | Model | Simple Arithmetic | Advanced Arithmetic | GLUE |
> | --- | --- | --- | --- |
> |OPT 125M | 10.4 | 0.6 | 1.1 |
> |OPT 350M | 10.9 | 0.6 | 1.2 |
> |OPT 1.3B | 13.1 | 0.5 | 1.8 |
> |OPT 2.7B | 14.4 | 0.6 | 2.1 |
> |OPT 6.7B | 16.8 | 0.5 | 2.7 |
> |OPT 13B | 17.4 | 0.6 | 2.9 |
>
> Note that for GLUE, we do not use a classifier and instead evaluate the generated output directly from the base model.
>
> Few-shot prompting of the base LLM further reveals minimal change from these results. We will provide these as baselines in the appendix.
>
> > Have the author tried different decoding settings/hypers.
>
> As the decoding setting itself is a tangential direction of research, we have not conducted a rigorous hyperparameter search along this direction.
>
> > Conduct experiments to control the imperfect-ness of tools. [...] Could the experiments include more complex tool-use settings?
>
> We thank the reviewer for the suggestions. We certainly agree with the reviewer that these would be beneficial in providing a more comprehensive study into our claims. However, as the reviewer acknowledges themselves, designing tools for these could be difficult, in particular if we want to continue evaluating within a continual learning scenario. In particular, our claim through our experimental observations is that tool-learning LLMs have the _*potential*_ to serve as better continual learners. While it is certainly true that our experiments are not exhaustive, nor do they cover the entire set of ways in which tools can be used, our results and the analysis that follow in section 6 explain in detail how and why we reach these conclusions.
>
> As such, we qualified all our claims as stating that tool-learning LLMs have the *potential* to become continual learners rather than outright stating this to be the case, as we acknowledge that our experiments are conducted in a simplified setting that might not extrapolate completely in the real world.
>
> -----
>
> ### References
>
> Schick et al., Toolformer: Language Models Can Teach Themselves to Use Tools, 2023
>
> -----
>
> Once again, we are grateful for the time the reviewer spent on providing constructive feedback to our work. We are hopeful that the provided clarifications are sufficient to address their questions while providing the changes that were requested.

---

> > ### Comment · Reviewer_GJtr · 2024-10-29
> > **Response to rebuttal**
> >
> > > rebuttal to “But, contrary to Figure 3…”;
> >
> > I think the experiments results is presented in a confusing way in that I have to jump to different figures (tools -> no tools, replay -> no replay, mixed dataset training -> continual setting) to understand what the authors are trying to get at. Even with the clarification in rebuttal, I only scratch the surface of what the authors are getting at. I suggest authors rethink the organization and presentation of the paper in future iterations.
> >
> >
> > > Author Response (2/N)
> >
> > Thanks for the clarification. This is helpful. It would be great to integrate the clarification to the paper in future iteration.
> >
> >
> > > rebuttal to “Why Q3 follows from main question”
> >
> > Thanks for the clarification. I do think this helps me understand the motivation. However, this also means that the authors didn’t set up the experiments properly to illustrate a clear argument that this is the case. What the authors responded in rebuttal seems to imply more than what’s written in the paper. The authors seem to be extrapolating too many arguments from a shared set of experiments. I suggest the author rethink the design of experiments to answer each research question in a more self-contained manner.
> >
> >
> > > decoding settings/hypers.
> >
> > I do think the authors should at least try some small-scale hyper parameter sweep with the base / few-shot prompting baseline for solidness, to make sure that the model really can’t accomplish the tasks and the gains from learning are better-supported.
> >
> >
> > > on the experiments for controlling imperfect-ness of tools
> >
> > I think the author should be rigorous about the claims / research made in the paper and; if there are claims that the authors are trying to make, such claims or research question should be accompanied by rigorous experiment designs.

---

> > > ### Author Response · Authors · 2024-11-02
> > > **Author Response**
> > >
> > > > I suggest authors rethink the organization and presentation of the paper in future iterations.
> > >
> > > We thank the reviewer for their suggestions. It appears that some of this confusion stems from perhaps the need for some more formal outlining of the terminology used in the paper. We are happy to provide this, in particular within Section 4 and 5 so that the terms we use are better explained and the different variables are more explicit.
> > >
> > > > It would be great to integrate the clarification to the paper in future iteration.
> > >
> > > We are glad this clarifies some points. We will certainly add this in future iterations.
> > >
> > > > However, this also means that the authors didn’t set up the experiments properly to illustrate a clear argument that this is the case. What the authors responded in rebuttal seems to imply more than what’s written in the paper. The authors seem to be extrapolating too many arguments from a shared set of experiments. I suggest the author rethink the design of experiments to answer each research question in a more self-contained manner.
> > >
> > > We are happy to narrow down our claims through adjusting the language if the author still believes this to be an issue. However, we still would like to make our case that the statements we make are supported.
> > >
> > > First, we state that learning through a tool-learning paradigm can potentially be beneficial for continual learning. We use a sequential task learning setup to show this, where we compare directly learning through natural language compared to learning to construct API calls that are executed with tools. Overall, our results show quite significant gains (provided some additional variables such as the use of experience replay).
> > >
> > > Second, we state in our conclusion that continual learning in this setting still isn't solved, which narrows down our claim. We only note that tool-learning shows a potential to help continual learning.
> > >
> > > Third, we state that tool-learning might be making better use of parametric knowledge, as the knowledge necessary to use them is more simple (from a human perspective) compared to needing to directly retrieve facts. This is supported by the increase in learning accuracy (both with and without replay), as it indicates that the model maintains greater flexibility for learning incoming tasks, which is directly linked to the use of parameters for storing knowledge.
> > >
> > > We hope these points clarify some of the concerns the reviewer might have with the scope of our claims and the support we provide for them.
> > >
> > > > I do think the authors should at least try some small-scale hyper parameter sweep with the base / few-shot prompting baseline for solidness, to make sure that the model really can’t accomplish the tasks and the gains from learning are better-supported.
> > >
> > > We remain grateful for the suggestion. Time-permitting, we are happy to add these details in a final version of the work. However, it is still tangentially related to the problem direction, and given the small deviations among seeds we believe it will not play a significant role within our problem setup. Nevertheless it is certainly something we will explore to provide more support to the claims.
> > >
> > > > I think the author should be rigorous about the claims / research made in the paper and; if there are claims that the authors are trying to make, such claims or research question should be accompanied by rigorous experiment designs.
> > >
> > > We appreciate this comment. We fully agree that claims that are made should be supported; we note above the claims of potential confusion that the reviewer might feel require some narrowing down.
> > >
> > > -----
> > >
> > > Once again, we are appreciative of the comments from the reviewer and have attempted to address them through this response. We hope some of these points better clarify our work and resolve remaining points of confusion. If the reviewer feels that this response along with our previous rebuttal provides answers to their remaining questions, we would be extremely grateful for their revised assessment of this work.

---

### Review · Reviewer_RACG · 2024-10-01

**Summary Of Contributions:**

This work investigates a continual learning scenario for tool learning by large language models (LLMs) assuming that a new tool, i.e., a new task, is continually appearing and thus LLMs are asked to acquire the knowledge for the new tool usage. Experiments are carried out on OPT language models by differentiating the size to measure the capacity with three task settings, toy arithmetic task, advanced arithmetic task and a subset of GLUE tasks. The former two relies on external tools that return accurate results, while the latter simply query an LLM to return an answer with potential errors. Experimental results show that LLMs can adapt to the new tools under the continual settings without forgetting prior knowledge.

**Audience:**

No

**Broader Impact Concerns:**

No statement exists for Broader Impact.

**Claims And Evidence:**

No

**Requested Changes:**

- This work needs further work on clarifying details in experimental setups, in particular, the definition of a task and how that is instantiated in the three experimental settings, i.e., toy, advanced and GLUE, as noted in my weakness comment. This work does not mention what LLM was used as a tool in the GLUE task.

- It is better to measure the impact of the order of tasks for continual learning.

- Rather than learning subtasks within a single setting, addition in toy arithmetic task, better to investigate more diverse tasks in continual learning, e.g., toy arithmetic, advanced arithmetic and GLUE, to see how those diverse tasks might affect each other or not.

**Strengths And Weaknesses:**

Strengths

-  The continual learning setting for tool usages sounds an interesting scenario and is probably practical in that new tools will emerge and LLMs need to adapt to such tools. Alternative is to directly let an LLM to given an answer, but the experiments show that adapting to a new tool is better than learning without a tool.

- The findings in the experiments share similar trends for other continual learning settings, e.g., learning new knowledge, though, this work is probably the first to investigate continual learning for tools and might have an impact to some researchers.

Weaknesses

- The important details of experimental setups are not clear. First of all, the definition for a task is not clear. If my understanding is correct, a single set of experiment, e.g., toy arithmetic task, has 4 tasks as noted in Table 1, and each task, e.g., "add" is treated as a task, but is it correct? A subset of GLUE is experimented, i.e., MNLI, QQP, SST-2 and CoLA, but did you treat each subtask as a task in continual learning?

- The impact of order is not clear. The motivation of this work is continual learning when facing a new task, but it is not clear the impact of learning addition first or division first. Given the different characteristics of continual learning, I feel better to investigate the order of the task.

- The three tasks are conducted separately, but I like to see what will happen when an LLM is asked to learn diverse tools, e.g., toy arithmetic, advanced arithmetic and GLUE. I think it is a more practical setting in that an LLM might need to learn a more advanced tools as the development of new technologies. I'm also interested in the advancements in tools employed in GLUE setting, e.g., more accurate tools are available for a certain task for some questions, but might fail to answer some particular questions.

---

> ### Author Response · Authors · 2024-10-08
> **Author Response (1/2)**
>
> We would like to thank the reviewer for their analysis of our work. We appreciate they mention they find the scenario interesting and holds some practical value in terms of how LLMs and tools work in the real world. We do appreciate that they mention this to be the first work to investigate continual learning for tools and that there may have impact for some researchers. We hope the responses we provide below can address the weaknesses mentioned and hopefully covers their requested changes.
>
> Please note that due to the direct relationship between some of the reviewer’s comments, we answer some of the listed points together in order to provide a more complete and comprehensive response.
>
> -----
>
> > The important details of experimental setups are not clear. First of all, the definition for a task is not clear. If my understanding is correct, a single set of experiment, e.g., toy arithmetic task, has 4 tasks as noted in Table 1, and each task, e.g., "add" is treated as a task, but is it correct? A subset of GLUE is experimented, i.e., MNLI, QQP, SST-2 and CoLA, but did you treat each subtask as a task in continual learning?
>
> Yes, this is correct.
>
> > This work needs further work on clarifying details in experimental setups, in particular, the definition of a task and how that is instantiated in the three experimental settings, i.e., toy, advanced and GLUE, as noted in my weakness comment. This work does not mention what LLM was used as a tool in the GLUE task.
>
> As mentioned above, each setting (toy arithmetic, advanced arithmetic, GLUE) is conducted on its own where the continual learning occurs along the different subtasks that are considered within the setting. The model is trained to learn one subtask after another while we evaluate performance on all tasks that were learned up to that point.
>
> Given that the tasks in the GLUE scenario are classification tasks, we use RoBERTa-base as a standard encoder-only model that can perform on each subtask. Each tool is a specific version of this model tuned on the relevant subtask so that it attains some reasonable level of performance (but is still imperfect).
>
> > The impact of order is not clear. The motivation of this work is continual learning when facing a new task, but it is not clear the impact of learning addition first or division first. Given the different characteristics of continual learning, I feel better to investigate the order of the task. [...] It is better to measure the impact of the order of tasks for continual learning.
>
> We have in fact already included this within our results, as the standard practice in continual learning is to vary along the order of tasks with the seeding. The effects can be viewed from our appendix, which includes results with standard deviations. These are very small relative to the raw accuracy, forgetting and learning accuracies that are observed. This suggests that the effects of shifting the task order is rather small and more consistent across orders.

---

> ### Author Response · Authors · 2024-10-08
> **Author Response (2/2)**
>
> > The three tasks are conducted separately, but I like to see what will happen when an LLM is asked to learn diverse tools, e.g., toy arithmetic, advanced arithmetic and GLUE. I think it is a more practical setting in that an LLM might need to learn a more advanced tools as the development of new technologies. I'm also interested in the advancements in tools employed in GLUE setting, e.g., more accurate tools are available for a certain task for some questions, but might fail to answer some particular questions.
>
> > Rather than learning subtasks within a single setting, addition in toy arithmetic task, better to investigate more diverse tasks in continual learning, e.g., toy arithmetic, advanced arithmetic and GLUE, to see how those diverse tasks might affect each other or not.
>
> We appreciate this comment and agree it would be a useful ablation. We provide a first investigation of results where we mix all tasks across all settings together. In particular, we mix all the subtasks from each setting, then randomly select an order to learn everything continually.
>
> | Model | Accuracy (W/O Tools) | Forgetting (W/O Tools) | Learning Accuracy (W/O Tools) | Accuracy (W/ Tools) | Forgetting (W/ Tools) | Learning Accuracy (W/ Tools) |
> | ------- |----------- | ------- | ------------- |----------- | ------- | ------------- |
> | OPT-125M | 11.2 | 68.8 | 31.2 | 23.6 | 91.2 | 95.4 |
> | OPT-350M | 12.7 | 71.2 | 35.2 | 25.1 | 93.7 | 95.8 |
> | OPT-1.3B | 24.3 | 64.6 | 61.4 | 27.1 | 93.6 | 95.7 |
> | OPT-2.7B | 22.2 | 62.8 | 63.2 | 26.2 | 92.4 | 96.1 |
> | OPT-6.7B | 25.2 | 72.5 | 65.8 | 24.8 | 92.6 | 96.3 |
> | OPT-13B | 24.5 | 64.8 | 67.4 | 26.2 | 92.1 | 96.6 |
> | LLaMA2-7B | 24.3 | 85.8 | 76.4 | 24.2 | 94.6 | 97.2 |
> | LLaMA2-13B | 25.4 | 83.3 | 75.3 | 24.8 | 91.3 | 98.1 |
> | LLaMA3-8B | 25.6 | 84.4 | 75.3 | 24.5 | 93.5 | 97.7
> | OPT-125M (+ER) | 38.3 | 10.2 | 43.1 | 93.2 | 1.1 | 96.1 |
> | OPT-350M (+ER) | 37.8 | 9.8 | 41.7 | 93.1 | 0.9 | 97.3 |
> | OPT-1.3B (+ER) | 59.7 | 7.8 | 61.3 | 94.5 | 0.7 | 97.5 |
> | OPT-2.7B (+ER) | 65.3 | 9.5 | 71.5 | 95.5 | 0.6 | 97.5 |
> | OPT-6.7B (+ER) | 69.6 | 8.3 | 74.4 | 96.2 | 0.4 | 98.1 |
> | OPT-13B (+ER) | 69.4 | 10.8 | 75.2 | 96.5 | 0.3 | 98.3 |
> | LLaMA2-7B (+ER) | 68.4 | 6.5 | 75.6 | 96.3 | 0.1 | 97.2 |
> | LLaMA2-13B (+ER) | 69.6 | 5.1 | 73.7 | 96.6 | 0.5 | 98.6 |
> | LLaMA3-8B (+ER) | 68.2 | 6.4 | 76.1 | 96.8 | 0.2 | 98.3 |
>
> These results follow our observations from using a single setting, showing that the same trends hold here. This serves as further justification regarding how the nature of the tasks is more or less independent from the difficulty of learning.
>
> -----
>
> Again, we would like to thank the reviewer for their comments and review. We greatly appreciate their acknowledgement of our work and topic, and we hope to have clarified any remaining concerns through this response.

---

### Decision · Action_Editor_FjYH · 2024-11-14

**Recommendation:** Reject

**Comment:**

This paper presents a study of continual learning in tool usage settings. The experiments explore two settings: arithmetic settings, where the "tools" used are perfect (implementing arithmetic operations as well as some more complex operations like GCD), and a GLUE setting where a PARAPHRASE tool is used.  Without replay, models tend to overfit the most recent tasks and forget how to use tools; with replay, this is ameliorated. The paper compares tool usage to non-tool usage methods of doing these tasks.

This paper covers some interesting territory. It is clearly written and the analysis is quite detailed. However, there are a few main weaknesses:

SIMPLICITY OF THE SETTING: zXRK (the most positive reviewer) and GJtr both have concerns about the simplicity of the setting. These results may not generalize to real tool usage scenarios. The main issue I see is that comparing tools against direct answering seems particularly idiosyncratic for these two tasks. Math is particularly dependent on the scale of the model and the amount of math data a model has been pre-trained on. I see these results more as telling us about the math capabilities of OPT rather than something broader. The GLUE setting is interesting, but also unique in that only one tool is used.

COMPARISON TO PAST WORK: Because of the simplicity of the tool usage scenarios here, I take reviewer zXRK's point about the comparison with Luo et al. I don't think this paper goes far enough beyond that one. The authors write:

> More specially, our interest is not in explicitly investigating forgetting, but instead into whether or not learning to use tools can in fact be beneficial for continual learning.

But because these settings are relatively simple and arithmetic essentially consists of classification and parsing, it's hard for me to see how we can compare tool use and non-tool use in an apples-to-apples way and draw generalizable conclusions.

PRESENTATION: As a more minor point, RACG raises presentational concerns which I echo, particularly about how the arithmetic tasks are structured. Table 1 doesn't quite make the training schedule as clear as it could be.

Overall, while the claims are supported in a very narrow sense, I think a paper like this is implicitly claiming to comment more broadly on the topic of continual learning and tool usage, and the reviewers and I feel it falls a bit short here. Exploring more sophisticated settings can differentiate it more strongly from Luo et al. And I think the story could be broadened by leveraging more challenging settings as well.

**Audience:**

Yes

**Claims And Evidence:**

No, I don't think so; see the comment below for a more thorough meta-review

**Resubmission Of Major Revision:**

The authors may consider submitting a major revision at a later time.